# Relation between total-column and near-surface NO$_2$ based on in-situ and PANDORA ground-based remote sensing observations

**Ying Zhang[a,b], Yuanyuan Wei[c*], Gerrit de Leeuw[a,d], Ouyang Liu[a,b], Yu Chen[e], Yang Lv[a], Yuanxun Zhang[b], Zhengqiang Li[a,b]**

[a] *State Key Laboratory of Remote Sensing and Digital Earth, Aerospace Information Research Institute, Chinese Academy of Sciences, Beijing, 100101, China*

[b] *University of Chinese Academy of Sciences, Beijing 100049, China*

[c] *Changcheng Institute of Metrology & Measurement, Beijing, 100095, China*

[d] *R&D Satellite Observations, Royal Netherlands Meteorological Institute (KNMI), 3730AE De Bilt, the Netherlands*

[e] *CMA Public Meteorological Service Centre, Beijing 100081, China*

*Yuanyuan Wei
Corresponding author at the Changcheng Institute of Metrology & Measurement.
Address: Huanshan Village 108, Haidian District, Beijing, China
Postcode:100095
Email:weiyy020@avic.com

**Abstract**

Nitrogen dioxide ($NO_2$) is a major pollutant which at high concentrations may affect human health. It is also a photochemically reactive gas which is important for the oxidation potential of the atmosphere and acts as a precursor for the formation of aerosol particles and ozone. However, monitoring of near-surface (NS) $NO_2$ faces the challenge of spatial discontinuity due to large distances between ground-based monitoring stations, whereas satellite remote sensing provides total vertical column density (VCD) which is related to near-surface (NS) concentrations in a complicated manner. In this study, the relation between total VCD and NS concentrations of $NO_2$ is analyzed based on total VCD from remote sensing observations using a ground-based Pandora spectrometer and NS $NO_2$ concentrations from in-situ observations. Both instruments were located at the Beijing-RADI site (Beijing, China) during January 2022. The ratio between total VCD and NS $NO_2$ concentrations varies throughout the day with substantially different relations in the morning and afternoon. During the night and morning, the atmosphere was vertically stratified, with disconnected layers which prevented vertical mixing of atmospheric constituents. In the afternoon, these layers connected allowing for vertical mixing and transport between the surface and the top of the boundary layer. Thus the prohibition of vertical transport in the morning and the mixing in the afternoon resulted in different relations between the total VCD and NS $NO_2$ concentrations. These different relationships have consequences for the use of satellite remote sensing to estimate NS $NO_2$ concentrations.

**Keywords**: Nitrogen dioxide, remote sensing, air pollution, Beijing, winter

## 1 Introduction

Nitrogen dioxide ($NO_2$) can have adverse effects on human health (Eum et al., 2022, 2019; Nordeide Kuiper et al., 2021; Kornartit et al., 2010). $NO_2$ plays an important role in atmospheric chemistry, and acts as a precursor for the formation of ozone and secondary aerosols in the atmosphere. The major sources of $NO_2$ are from fossil fuel burning such as power plants, traffic and households. Because of these anthropogenic sources, together with the relatively short atmospheric lifetime of $NO_2$, high tropospheric $NO_2$ concentrations are usually observed near highly industrialized regions (van der A et al., 2006), densely populated agglomerations (de Souza et al., 2022) and power plants (Tang et al., 2024), as well as along major highways (Goldberg et al., 2021) and shipping lanes (Ding et al., 2018). In addition, $NO_2$ is produced from some natural sources such as lightning and soil emissions.

Concentrations of $NO_2$ in the atmosphere can be measured using satellite-based sensors providing total and tropospheric column densities, ground-based remote sensing using MAX-DOAS or Pandora instruments, or in situ instruments. Satellite remote sensing is currently a widely used technique, for example using the Ozone Monitoring Instrument (OMI, Levelt et al., 2006) on the Aura satellite, and the TROPOspheric Monitoring Instrument (TROPOMI, Veefkind et al., 2012) on the Sentinel-5 Precursor (S5P) satellite. Satellite data show that the total vertical column density (VCD) of $NO_2$ is highly variable in space and time (e.g., Lamsal et al., 2014; Fan et al., 2021). Duncan et al. (2016) analyzed global $NO_2$ observed by OMI from 2005-2014 and found that $NO_2$ levels were initially high over China but had significantly decreased over the Beijing, Shanghai and the Pearl River Delta (PRD) regions in 2014, in response to pollution control measures. In particular, over the PRD region the $NO_2$ concentrations decreased by about 40%. Also in the following years, the $NO_2$ concentrations over China have substantially diminished in response to the implementation of emission reduction policy (e.g. van der A et al., 2017; Fan et al., 2021; de Leeuw et al., 2021) and fell below the 2008 level in 2017 (Zhao et al., 2023). However, the decrease seems to have flattened in recent years (Fan et al., 2021).

The Pandonia Global Network (PGN) of Pandora Spectrometer Instruments has
been established in 2018 (http://www.pandonia-global-network.org/, last accessed:
10th July 2024) to provide "quality observations of total column and vertically resolved
concentrations of a range of trace gases". The PGN data are used, for instance, for the
validation of products from environmental satellites. However, the comparison of OMI
total VCD of $NO_2$ with Pandora observations at 6 sites in Korea and the USA by
Herman et al. (2019) showed that mean and daily Pandora $NO_2$ concentrations were 50%
or more higher than those retrieved from OMI at sites that were frequently contaminated,
such as Seoul, Busan and Washington DC. Tzortziou et al. (2018) reported that Pandora
total VCD of $NO_2$ during the KORUS-AQ coastal cruise experiment (Tzortziou, et al.,
2015) were 10-50% higher than OMI-derived total VCD of $NO_2$. The relationship
between total VCD and NS concentrations of $NO_2$ was complex from the analysis of
data from the DISCOVER-AQ campaign in the Baltimore-Washington region in July
2011; the discrepancies were suggested to be caused by the large field of view of OMI
(Flynn et al., 2014; Knepp et al., 2015; Reed et al., 2015; Tzortziou et al., 2015).
Preliminary validation of total VCD of $NO_2$ from the Ozone Mapping and Profiler Suite
(OMPS) aboard the joint NASA/NOAA Suomi National Polar-orbiting Partnership
(Suomi NPP) satellite by Huang et al. (2022) in the USA showed that OMPS total VCD
of $NO_2$ tends to be lower in polluted urban areas and higher in clean areas/events than
Pandora observations. Ialongo et al. (2020) and Zhao et al. (2020) obtained similar
results from the validation of TROPOMI total VCD of $NO_2$ using PGN data but the
differences were significantly smaller than for the OMI and OMPS data with a coarser
spatial resolution than TROPOMI. The validation of TROPOMI total VCD versus
Pandora data at the Beijing-RADI site shows the good performance of TROPOMI (Liu
et al., 2024). It is noted that Liu et al. re-sampled the TROPOMI data to a spatial
resolution of $100 \times 100$ m$^2$, i.e. similar to that of the Pandora observation area.
Satellite-derived total VCD of $NO_2$ data are often used to determine trends (e.g.,
van der A et al., 2017; Fan et al., 2021) but, in view of the above, the relation between
total VCD and NS concentrations of $NO_2$ is more complex. For instance, Fan et al.

(2021) discussed the total VCD/NS relationship for selected major urban regions in China during the first 20 weeks after the COVID-19 lockdown and observed substantial differences (their Figure 9). Chang et al. (2022) analyzed data from the Geostationary Environment Monitoring Spectrometer (GEMS) Map of Air Pollution (GMAP) campaign conducted during 2020–2021. Their results indicate that total VCD and NS concentrations of $NO_2$ exhibit a stronger correlation under advective boundary layer conditions at high wind speeds, where the vertical distribution of $NO_2$ is more uniform. In contrast, in the presence of plumes from large point sources, either decoupled from the surface or transported from nearby cities, enhance the vertical heterogeneity of $NO_2$. These plumes contribute to a less consistent relationship between total VCD and NS concentrations of $NO_2$. Similarly, Liu et al. (2024) show different relations between total VCD and NS concentrations of $NO_2$ for low and high concentrations which are qualitatively explained in terms of transport and local emissions. Moreover, Thompson et al. (2019), using data from the KORUS-AQ coastal cruise experiment, reported that there is no consistent correlation between total VCD and NS concentrations of $NO_2$ across different cases and that the relation between total VCD and NS concentrations of $NO_2$ is complex. Thus, to accurately assess $NO_2$ pollution in China and effects on air quality, accurate ground-based observations are needed.

Although a large number of ground-based $NO_2$ observation stations have been established in China since 2012 by the China National Environmental Monitoring Center (CNEMC) of the Ministry of Ecology and Environment of China (MEE) for the provision of the ground-based monitoring data (available at http://www.mee.gov.cn/; last access: 08 July 2024), there are still large areas for which no data are available. Satellite data can fill these gaps by converting satellite observations of aerosols and trace gases from total VCD and NS concentrations. Such data are usually provided from sensors flying on polar-orbiting satellites with global coverage but with a single overpass per day which at most latitudes cannot provide the daily variability of $NO_2$ characteristics. However, with the launch of geostationary satellites, spatial and temporal distributions of $NO_2$ concentrations can be obtained within the satellite field

of view throughout the day. The GEO-KOMPSAT-2B geostationary satellite, launched by the National Institute of Environmental Research (NIER) under the Ministry of Environment, Korea, in February 2020, carries the Geostationary Environment Monitoring Spectrometer (GEMS), which provides high-resolution measurements of total VCD of key air quality components (Kim et al., 2020). With the launch of GEMS, the Asian region was the first to achieve coordinated hour-by-hour monitoring of pollutants. GEMS will form a constellation of satellites to monitor air quality globally with high temporal and spatial resolution, together with the Tropospheric Emissions: Monitoring Pollution (TEMPO) mission, launched by NASA on 7 April 2023 to cover the North American region (https://tempo.si.edu/overview.html) and Sentinel-4 planned to be launched on the Meteosat Third Generation Sounder (MTG-S) by the European Organisation for the Exploitation of Meteorological Satellites (EUMETSAT) in 2025 (https://www.eumetsat.int/meteosat-third-generation-sounder-1-and-copernicus-sentinel-4, last visited 18 April 2025).

While geostationary satellites enable continuous daytime observations of total VCD of $NO_2$, discrepancies between total VCD and NS concentrations of $NO_2$ concentrations remain a critical challenge. The weak correlation between NS $NO_2$ concentrations and satellite-derived total VCD of $NO_2$ (Lamsal et al., 2014) is closely tied to differences in their vertical distribution, atmospheric lifetimes, and chemical reaction pathways (Xing et al., 2017). Despite extensive efforts to derive NS $NO_2$ concentrations from total VCD measurements (Chang et al., 2025; Wei et al., 2022; Zhang et al., 2022; Dou et al., 2021), the dynamic complexity of the planetary boundary layer introduces substantial uncertainties. Moreover, prior studies have emphasized the roles of vertical $NO_2$ distribution (Sun et al., 2023; Zhang et al., 2023; Kang et al., 2021) and regional pollutant transport contributions (Yin et al., 2025; Dong et al., 2020; Chang et al., 2019; Li et al., 2017), but research explicitly linking regional transport processes to vertical $NO_2$ concentration gradients and elucidating their interactive effects remains limited. Song et al. (2024) obtained NS $NO_2$ concentrations based on the Himawari-8 geostationary satellite using machine learning, which has good performance in the noon

and afternoon, and relatively poor performance in the morning. These knowledge gaps are further exacerbated by satellite data limitations in resolving NS pollution, which has direct implications for human health assessments. To address these challenges, we need to integrate ground-based remote sensing observations with in situ NS $NO_2$ measurements to investigate vertical decoupling phenomena, and investigate the influence of distinct pollutant transport pathways on NS $NO_2$ pollution dynamics.

The first operational Pandora instrument in China has been installed at the Beijing-RADI site in 2021, for the ground-based remote sensing of several trace gases, including $NO_2$. The Beijing-RADI Pandora instrument is part of the PGN network and all data are publicly available via the PGN website (https://data.pandonia-global-network.org/Beijing-RADI/Pandora171s1/, last accessed: 22 Jan 2025) within one day of the observations. The aim of this study is to analyze the relationship between total VCD of $NO_2$ obtained by remote sensing and NS measurements during a field experiment at the Beijing-RADI site during January 10-29, 2022. In order to better determine and understand the relation between total VCD and NS concentrations of $NO_2$ concentrations, we also used auxiliary data such as simultaneous measurements of $PM_{2.5}$ mass concentrations, lidar observations, meteorological parameters, and satellite observations. Section 2 presents the experiments and data, Section 3 presents the main results, and the conclusions and discussion are presented in Section 4.

## 2 Materials and Method

### 2.1 Site description

Beijing is a metropolis with an area of 16,800 square kilometers and a population of nearly 22 million (2023) (https://worldpopulationreview.com/world-cities/beijing-population, last accessed: 22 Jan 2025). High mountains are located to the north and west of Beijing (Fig. 1). The rapid economic development of Beijing and the topography of the area lead to emission of pollutants which may either disperse or accumulate, depending on wind direction and wind speed. During northwesterly winds, clean air is transported from the mountains, whereas during southerly winds polluted

air is transported from the highly industrialized North China Plain. The southerly airflow is blocked by the mountains to the west and north and thus pollution accumulates, in particular during certain weather conditions conducive for the formation of smog, such as low wind speed. Photochemical processes may further contribute to the build-up of pollution which may result in the formation of haze.

The Beijing_RADI site is located at the roof (22 m above the surface) of the Aerospace Information Research institute of the Chinese Academy of Sciences (40.004° N, 116.379° E, elevation 59 m), which in turn is located in the north of Beijing between the Fourth and Fifth Ring Roads at the edge of the Olympic Parc. The site is representative for an urban background affected by vehicle exhaust, combustion and domestic emissions including those from heating during wintertime.

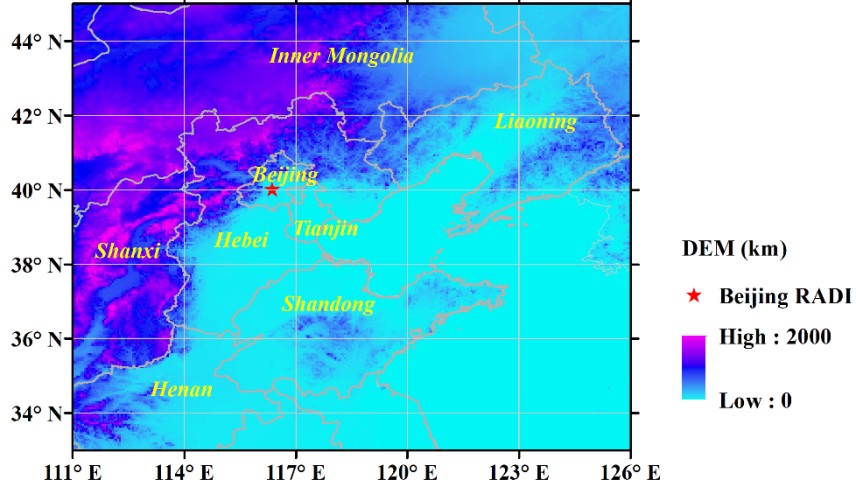

**Figure 1**. Digital elevation map of the study area showing the location of the Beijing-RADI site (40.004° N, 116.379° E, altitude at 59 m) (red star) and surrounding mountains.

A short-term field experiment was carried out at the Beijing-RADI long-term observation site during January 10-29, 2022. Pandora provided total VCD of $NO_2$ concentrations in 4 layers, information on the spatial distribution of total VCD of $NO_2$ was obtained from TROPOMI and a lidar provided aerosol backscatter profiles showing the vertical structure and evolution of the atmospheric boundary layer. The NS parameters were measured with instrumentation mounted near the Pandora on the roof

of the AirCAS building, as described in the following sections.

## 2.2 Observations of Column-Integrated Parameters and Vertical Profiles

### 2.2.1 Pandora

Pandora is a UV-visible spectrometer which can provide high-quality
measurements of spectrally resolved direct-sun/lunar or sky scan radiances. It uses
direct solar measurements to obtain total VCD of $NO_2$, and sky measurements to obtain
the vertical layer concentrations of $NO_2$, with a FOV of 2.6° in direct sun mode and
1.5° in sky mode (Cede, 2024). Based on the Beer-Bouguer-Lambert law, the spectra
observed at 400~470 nm in direct-sun mode are used to invert total VCD of $NO_2$ using
the differential optical absorption spectroscopy (DOAS) technique of trace gas spectral
fitting. Pandora's direct sun measurements depend only on the geographic location with
a known solar zenith angle which simplifies the air mass factor for correction of the
atmospheric light path (Chang et al., 2022). Pandora measures total VCD of $NO_2$ with
a clear-sky precision of 0.01 DU and a nominal accuracy of 0.1 DU (Herman et al.,
2009). In view of this high precision, we use total VCD of $NO_2$ from the nvs3 product
in this study and select data with quality control flag of L10. Diffuse (scattered)
radiation is measured at 5 pointing zenith angles (PZAs) in sky mode which, together
with the direct sun measurement, provides information on the tropospheric VCD and
on the surface concentrations. The PZAs are 0°, 60°, 75°, 88° and a maximum angle
taken as 89°. The measurements are taken in a V shape (all angles are measured twice
around a central angle) as described in Cede (2024). Four partial columns of $NO_2$
concentrations are provided by the PANDORA inversion. The first step is the estimation
of the effective height corresponding to a given PZA, and then calculate differential air
mass factors for the $NO_2$ and the air-gas for each layer. The profile shape of the partial
columns is determined as a variation of the air-gas shape. The average number density
of the $NO_2$ in each layer is then calculated. The partial column amounts can be obtained
from the concentrations multiplied with the layer width as described in the Manual for
Blick Software Suite (Cede, 2024), Section 6.7. The $NO_2$ of the partial column can be
obtained from the uvh3 product which was downloaded from the PGN website
(https://pandonia-global-network.org, last accessed: 22 Jan 2025). We converted these
partial column concentrations into layer-averaged volume mixing ratios and
interpolated them to 6 standard levels (0.2, 0.5, 1.0, 1.5, 2.0, 2.5 km) for visualization.
**2.2.2 Lidar**
A small lidar developed by the Hefei Institute of Physical Sciences, Chinese
Academy of Sciences, was used for continuous measurements of aerosol backscatter
profiles during day and night. The GBQ L-01 aerosol lidar consists of a laser, optical
unit, control unit board, high-speed signal acquisition card, industrial motherboard and
communication module. The GBQ L-01 aerosol lidar uses a high-frequency pulse laser
emitting linearly polarized light at a wavelength of 1064nm. The optical unit consists
of a transmitter and a receiver. The optical transmitter unit emits laser light pulses,
which are expanded before they are emitted into the atmosphere. The optical receiver
unit consists of a telescope which focuses the back-scattered light onto an optical
detector which in turn is connected to an amplifier unit. The vertical and parallel
polarized components of the back-scattered light are separated by the polarizing prism
in the receiving channel. The industrial motherboard carries lidar acquisition and
control software and data analysis software to control the overall operation of the
system.
**2.2.3 TROPOMI**
The TROPOMI (TROPOspheric Monitoring Instrument) is a passive-sensing
hyperspectral nadir-viewing imager aboard the Sentinel-5 Precursor (S5P) satellite,
launched on 13 October 2017. S5P flies at an altitude of 817 km in a near-polar sun-
synchronous orbit. The local equator overpass time in the ascending node is 13:30, and
the repetition period is 17 days (KNMI, 2017). TROPOMI's four separate
spectrometers cover wavelengths in the ultraviolet (UV), UV–visible (UV-VIS), near-
infrared (NIR) and short wavelength infrared (SWIR) spectral bands (Veefkind et al.,
2012). The $NO_2$ used in this study is derived from spectral measurements of solar
radiation in TROPOMI's UV–VIS wavelength bands (van Geffen et al., 2015, 2019).
Compared to the relatively small field-of-view of the Pandora instrument, the size of
the TROPOMI ground pixel ($3.5$ km $\times$ $5.5$ km; across $\times$ along track) is relatively large.
In addition, only tropospheric vertical column densities (VCDs) for $NO_2$ were available
during the time period of this study. The TROPOMI tropospheric VCDs for $NO_2$ are
only used as a qualitative reference for upwind concentrations for the evaluation of
effects of long-range transport using air mass backward trajectories and not for
quantitative analysis. Furthermore, tropospheric $NO_2$ column densities are used
because these are more representative of near-surface $NO_2$.
**2.3 Near Surface Measurement**
**2.3.1 Trace Gas Analyzer**
The Thermo Fisher Scientific Model 42i Trace Level Chemiluminescence NO-
$NO_2$-$NO_x$ Analyzer was used to measure NS $NO_2$ concentrations. This instrument first
transforms $NO_2$ into nitric oxide (NO) using a molybdenum $NO_2$ to NO converter
heated to about 325 °C. Then, NO and ozone ($O_3$) react to produce a characteristic
luminescence with an intensity linearly proportional to the NO concentration (Model
42i Trace Level Manual, 2007). $NO_2$ values are derived by subtracting NO from $NO_x$
measurements. Measurements were made every minute during the observation period.
**2.3.2 Beta attenuation monitor**
Ground-based near-surface $PM_{2.5}$ concentrations were measured using the beta
attenuation monitor Met One BAM-1020 (BAM 1020 particulate monitor operation
manual) equipped with a $PM_{2.5}$ inlet. The Met One BAM-1020 collects aerosol particles
on glass filter tape. $PM_{2.5}$ is measured using beta rays generated by a small $^{14}C$ source
(https://metone.com/products/bam-1020/). At the start of every measurement cycle, the
flux of beta rays is measured across clean filter tape, to determine a zero reading. Next,
the filter tape is advanced and ambient air is sampled at the same spot, with a controlled
air flow, thereby impregnating the tape with $PM_{2.5}$. After the sampling is completed, the
tape retracts and $PM_{2.5}$ samples are dried (in an environment with relative humidity
lower than 40% which removes most of the water content) by a built-in heater. Then
the concentration of $PM_{2.5}$ collected on the filter tape is measured as described above.
Samples are taken every hour.

### 2.3.3 Auxiliary meteorological data

In addition to the above observations, we also use weather maps, meteorological surface observations and sounding observations published by the World Meteorological Organization (WMO) to aid in our analyses. Weather maps for the Asian region are published by the Korea Meteorological Administration and can be downloaded at http://222.195.136.24/chart/kma/data_keep (last accessed: 22 Jan 2025). We downloaded the surface and sounding observations of meteorological station 54511 in Beijing, located at 39.93N, 116.28E, which is part of the WMO network. These data are available from the website of the University of Wyoming (http://weather.uwyo.edu/surface/) (last accessed: 22 Jan 2025). Although this station is far away from our experimental site (about 23 km), it is representative of the macroscopic changes of the meteorological conditions in Beijing.

### 2.4 HYSPLIT Model

To better understand the regional transport pathways and source regions at different altitudes, backward trajectories from the Hybrid Single Particle Lagrangian Integrated Trajectory (HYSPLIT; Draxler & Hess, 1998) model were used. The HYSPLIT model assumes that the parcel trajectory is formed through time integration and spatial differences when moving in the wind field. The path of the air mass is mainly related to the air flow situation, pressure system movement and topography (Draxler & Hess, 1998). The HYSPLIT model has the ability to deal with a variety of meteorological input fields and physical processes, and can also be used to describe atmospheric transport, diffusion and deposition of pollutants and harmful substances (Stein et al., 2015). In this study, the backward trajectories were initialized for arrival at the Beijing-RADI site at 300 m, 500 m, and 1000 m. The HYSPLIT model was run at https://www.ready.noaa.gov/HYSPLIT_traj.php, with the input meteorological field data ($0.25° \times 0.25°$) provided by the Global Forecasting System.

## 3 Results and Analysis

### 3.1 Data overview

Time series of the measured NS and total VCD of $NO_2$ during the study period are presented in Fig. 2a. For comparison of NS $NO_2$ concentrations with Pandora observations, they are expressed in mg m$^{-3}$. Fig 2a shows the common diurnal variation of the $NO_2$ concentrations, i.e. a gradual decrease in the morning to a minimum around mid-day, followed by a gradual increase in the afternoon to a maximum value during the night. This diurnal variation is due to photochemical reactions during daytime, meteorological effects and anthropogenic emissions during certain hours (for instance during rush hour) (e.g., Atkinson, 2000; Boersma et al., 2009; Zhang et al., 2016; Cheng et al., 2018; Li et al., 2021). Total VCD of $NO_2$ concentrations can only be measured with Pandora during day time. The diurnal variations between 8:00 and 16:00 local Beijing time (UTC+8; throughout this paper local time, LT, will be used) at the Beijing-RADI side are similar to those of NS $NO_2$. Based on the variation of the NS $NO_2$ concentrations (Fig. 2a) three periods are considered during the study period: Period I: 10 to 18 January, with strong diurnal variations and high $NO_2$ concentration peaks; Period II: 19 to 24 January, NS $NO_2$ sharply decreases and then increases with stronger fluctuations, but total VCD of $NO_2$ are not available due to the presence of clouds; Period III: 25 to 30 January, a sudden drop occurred on January 25, and low $NO_2$ concentrations with some narrow peaks lasted until the 30[th].

The time series of the NS PM$_{2.5}$ concentrations in Fig. 2b shows four peaks in Period I, with maximum values during the night and very low concentrations (<10 μg m$^{-3}$) during daytime. The maxima were relatively low on 12 and 17 January (~25 μg m$^{-3}$), whereas on 14 and 18 January the PM$_{2.5}$ peak concentrations were ~120 and ~70 μg m$^{-3}$. During Period II, the PM$_{2.5}$ concentration increased steadily from less than 25 μg m$^{-3}$ on 20 January to more than 125 μg m$^{-3}$ on the 24[th], with similar day/night variations as in Period I. During Period III, the PM$_{2.5}$ concentrations were relatively low (<25 μg m$^{-3}$) and there was no clear diurnal variation.

The air temperature and relative humidity (RH) during the study period are
presented in Fig. 2c and the wind speed and wind direction are presented in Fig. 2d.
During Period I, air temperature, RH and wind speed all varied strongly with a clear
diurnal pattern: elevated wind speed during the day, with daily maxima between about
7 and 13 m s$^{-1}$, and very low wind speed during the night ($<2$ m s$^{-1}$); day time air
temperatures around 0$^{o}$C and night time temperatures around -10$^{o}$C; dry air during the
day (RH~20%) and more humid during the night (RH of 60-80%). The wind was mostly
from northerly directions (NW-NE) and veering during the night. During Period II, the
air temperature increased gradually from about -5$^{o}$C to about 0$^{o}$C, with small diurnal
variations, and RH increased initially from 40% to 60% on 20 January and then
gradually to about 70%, with little day/night variations. During the nights of 22-25
January the humidity was very high, the RH sensor saturated and reported maxima close
to 100%. Wind speed during this period was low ($< 3$ m s$^{-1}$) and wind direction was
mostly SE. During period III, day/night temperature and RH fluctuations occurred, with
day time air temperatures above 0$^{o}$C and gradually rising and RH varying between 20%
during the day and 60% during the night. Wind speeds were higher during the day,
mostly a little higher than 3 m s$^{-1}$, than during the night (close to 0 m s$^{-1}$) and wind
direction was mostly northerly.
The variations of the NO$_2$ and PM$_{2.5}$ concentrations were similar in the sense that
the minimum and maximum peak concentrations occurred at about the same time, but
with differences in the ratios between minima and maxima. The occurrence of peak
concentrations during the night is consistent with the variation in meteorological
conditions, with maxima during low wind speed and low air temperature, conducive for
the formation of a nocturnal boundary layer in which the concentrations accumulate
near the surface. This is observed during Periods I and III. During Period II, however,
there were no Pandora observations during day time due the occurrence of clouds. This
suggests that clouds may also have been present during the night. Hence radiative
cooling was reduced and air temperature did not decrease as much as during the other
periods. Wind speed was low, thus pollutants were not transported away and
accumulated in the area, as also indicated by the high RH. Hence the concentrations of
$NO_2$ and $PM_{2.5}$, as well as RH, gradually increased during period II, with relatively
small diurnal variations.

383        For further analysis, we selected two cases for which both total VCD and NS

concentrations of $NO_2$ data were available, including lidar observations and air mass
trajectories. The selected cases are the periods when both $NO_2$ and $PM_{2.5}$ concentrations
were high, i.e. on 14 and 18 January, when the 24-hour average $NO_2$ concentrations
exceeded 80 mg m$^{-3}$. The first pollution episode (case 1) started in the afternoon of
January 14 and ended in the morning of January 15. The other pollution episode with
high NS $NO_2$ occurred on January 18 (case 2). The diurnal variation of the NS $NO_2$
concentrations was similar in both cases, while the air temperature, RH, wind speed
and wind direction show that also the meteorological situations were similar. However,
differences are observed in the temporal variations of total VCD of $NO_2$ concentrations
versus NS $NO_2$ concentrations and in $PM_{2.5}$ concentrations.

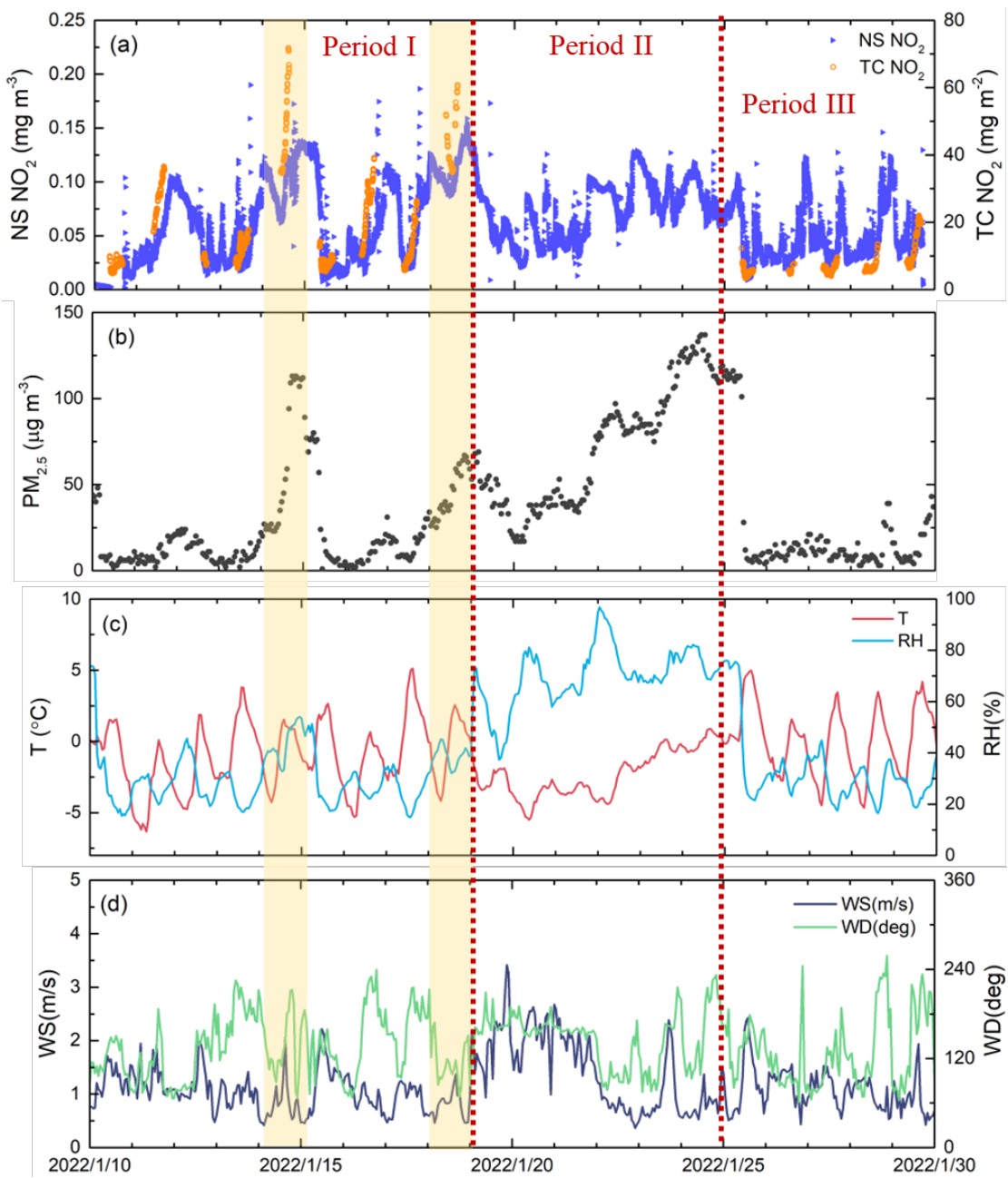

**Figure 2**. Time series of observed parameters from Jan 10 to 29, 2022 (a) total VCD and NS concentrations of $NO_2$ concentrations, (b) NS $PM_{2.5}$ concentration, (c) temperature and related humidity, and (d) wind speed and wind direction from WMO meteorological station 54511 in Beijing. The vertical dotted lines mark the boundaries between the three periods and the yellow shaded rectangles mark the two cases discussed in Section 3.2.

**3.2 Variations of total VCD and NS concentrations of NO$_2$ during the two selected cases**

Many studies (Yin et al., 2025; Dong et al., 2020; Chang et al., 2019; Li et al., 2017) indicated that the pollution in Beijing mainly originates from three pollution transport patterns: southwest (SW), southeast (SE), and south mixing (SM) patterns. Among them, the most representative are the SW transport path from Shanxi province to Shijiazhuang-Baoding-Beijing and the SE transport path from Shandong province to Cangzhou-Langfang-Tianjin-Beijing. However, few studies have explored the vertical transport of pollution types. Cases 1 and 2 are representative for the widespread SW and SE patterns, respectively. The processes influencing the concentrations are identified based on lidar data, providing information on the boundary layer structure, together with large scale weather maps and air mass trajectory analyses, providing information on sources of pollutants and their transport over a wider area. Our study primarily focuses on the correlation between total VCD and NS concentrations of NO$_2$. Total VCD of NO$_2$ are provided by the passive remote sensing instrument PANDORA, which does not provide reliable observations on cloudy days, as mentioned above. This is the reason why we only selected cases during Period I and did not obtain cases during Period II, when pollution was more severe.

**3.2.1 Case 1: Disconnected boundary layers merging (14 January, 2022)**

Time series of the total VCD and NS concentrations of NO$_2$ on 14 January 2022 (Fig. 3a) show their different evolution throughout the day. The NS NO$_2$ concentrations are available for every minute during the whole day and show a gradual decrease from about 0.11 mg m$^{-3}$ between midnight and 04:00 to about 0.065 mg m$^{-3}$ at 10:30. After 10:30 the concentrations increased to 0.11 mg m$^{-3}$ at 13:30 and hardly changed until about 16:00 after which they strongly fluctuated (0.04-0.175 mg m$^{-3}$) and then reached a steady value of about 0.12 mg m$^{-3}$ from 19:00 till midnight. The strong fluctuations may have been caused by emissions during evening rush hour, domestic heating and other activities producing NO$_2$, followed by stabilization during the evening.

Pandora uses direct sun observations and during this campaign in the winter time,
high quality total VCD and NS concentrations of $NO_2$ are only available between 10:30
and 15:30. The data in Fig. 3a show initially a similar behavior of total VCD and NS
concentrations of $NO_2$, with little variation between 10:30 and 11:30. Thereafter, both
NS concentrations and total VCD increased, initially slower for the total VCD than for
the NS concentrations. After 13:00 the NS concentrations levelled off while the total
VCD increased much faster. Between 12:00 and 15:00 the total VCD increased from
$40 \, \text{mg m}^{-2}$ to $72 \, \text{mg m}^{-2}$, almost a doubling, then decreased to $64 \, \text{mg m}^{-2}$. The difference
in the temporal behavior between the total VCD and NS concentrations of $NO_2$ is
amplified in Fig 3b which shows a scatterplot between the total VCD and NS
concentrations. Observations before and after 13:00 are plotted with different symbols
and color coded in blue and red, respectively. For each of these two data sets, before
and after 13:00, TS and NS concentrations are well correlated with linear correlation
coefficients R of 0.94 and 0.85, respectively, but with significantly different slopes.
The different behavior of the total VCD and NS concentrations of $NO_2$ can be
explained by considering the dynamical behavior of the boundary layer structure. Lidar
observations reveal the vertical structure of the atmospheric boundary layer from the
variation of the lidar signal as a function of height. A 3-D plot of the vertical variation
of the lidar signal, measured on 14 January 2022 at the Beijing-RADI site, close to the
Pandora and the ground-based measurements, is presented in Fig. 3c. The lidar signals
are color-coded according to the scale to the right of Fig. 3c and each vertical line shows
the variation of the lidar signal with height, plotted along the primary vertical axis. The
time of measurement of each profile is plotted along the horizontal axis. The lidar signal
in this figure is range-corrected, i.e. corrected for attenuation as the laser light
propagates in the atmosphere away from the emitter and, after backscattering by aerosol
particles, back to the receiver. The time between emission of the laser pulse and
receiving the backscattered signal is a measure for the height where the backscattering
takes place (after correction of the slant to a vertical optical path) and the intensity is a
measure for the aerosol concentration. This is illustrated with the data in Fig 3c. For

example, the data show an aerosol layer between 08:00 and 13:00, located at a height between about 800 and 900 m, as indicated by the large lidar signal (yellow and red, i.e., between about 1.2 and 2.2), with light blue above and below, indicating lower aerosol concentrations. Between about 400 and 500 m a dark blue area can be observed, which indicates very low backscatter and thus the absence of aerosol, whereas further down toward the surface, backscatter is observed with a varying intensity. The vertical variation of the lidar signal, i.e. indicating the presence of aerosol in the layer adjacent to the surface up to about 400 m, a layer with no aerosol between 400 and 500m and an elevated intense aerosol layer above, indicates a situation of a disconnected boundary structure with two layers which are not connected and thus no material can be exchanged between these layers. Such a situation can occur due to nocturnal cooling when the surface is cold due to radiation cooling and cools the layer adjacent to the surface (Stull, 1988). In this layer, no mixing occurs and material emitted near the surface accumulates. The atmospheric trace gases and aerosol in the warmer layer above are trapped in that layer and exchange with the cold layer below is prohibited due to the temperature gradient. Hence the two layers become disconnect and may separate.

The occurrence of such a situation is consistent with the observations discussed in Section 3.1 and Fig. 2, with low wind speed, lowest air temperature during period I (-12°C) and enhanced RH (indicating trapping of water vapor together with decreased air temperature). Also the lidar data in Fig. 3c indicate the occurrence of such a situation, with a well-mixed shallow boundary layer between midnight and 03:00, an indication of an internal boundary layer starting to form after about 04:00, disconnected from the layer above. The internal boundary layer rises gradually until about 11:00, with the clean layer above, and a new layer appears around 07:00, probably due to advection. Note that wind direction was south-easterly during a short period of time on 14 January with a wind speed of 2 m s$^{-1}$, slightly more than during the rest of the day when the wind direction was northerly. During south-easterly winds, polluted air may be advected to the Beijing-RADI site, whereas during northerly wind clean air is advected

(Liu et al, 2024).
From 12:00, the lower layer deepened and backscatter is observed from the clean
layer indicating that aerosol is gradually mixed into that layer which completely
disappears around 14:00. At the same time, the lidar signal from the growing lower
layer increases gradually whereas after 13:00 the lidar signal from the upper layer
becomes smaller, indicating that the aerosol concentration becomes lower until both
layers are mixed around 14:00 into a well-mixed boundary layer. After 15:00, the lidar
signal increases, first near the surface and then growing throughout the boundary layer.
The increase of the NS concentrations is consistent with the highest $PM_{2.5}$
concentrations as presented in Fig. 2b and the overall increase of the lidar signal,
indicating increasing aerosol concentrations. This is confirmed by AERONET AOD
observations at the Beijing-RADI site (https://aeronet.gsfc.nasa.gov/cgi-
bin/data_display_aod_v3?site=Beijing_RADI&nachal=2&level=2&place_code=10)
which however were only available until 16:00 LT.
The vertical variation of the $NO_2$ concentrations, derived from the Pandora sky
radiance measurements at four elevations, is presented in Fig. 3d. Pandora data are only
available during day time and therefore only the period from 08:00-16:30 can be shown.
The $NO_2$ concentrations are available in 4 layers. Assuming that $NO_2$ is uniformly
distributed within each layer, the data were interpolated to form a time series of $NO_2$
vertical distributions, similar to the lidar profiles. The data in Fig. 3d show the similar
behavior of the $NO_2$ concentrations and the aerosol backscatter, with increasing
concentrations between 12:00 and 17:00 and their vertical mixing. In particular the
increase around 15:00 is evident in both the Pandora and lidar observations. However,
the Pandora observations do not show the occurrence of disconnected boundary layers
in the morning. Instead, the Pandora observations show an enhanced layer between 300-
800 m, rather than the more detailed structure visible in the lidar data. These differences
stem from the difference in vertical resolution between the PANDORA and the lidar:
the total VCD from PANDORA is divided into two layers (approximately 300-800 m
and 800-1700 m) within the detailed stratified height range (300-1000 m) observed by
lidar. Consequently, the fine stratified structure within 1000 m cannot be identified with
the available PANDORA data.
The overall similarity between the variations in the lidar and Pandora observations
supports the use of lidar observations to explain the dynamic behavior of the $NO_2$
concentrations. In particular, the different relations between total VCD and NS
observations before and after 13:00 (Fig. 3b), can be explained by the occurrence of
disconnected layers. The variations of the NS $NO_2$ concentrations until 13:00 reflect
the effects of chemical processes and emissions within the atmospheric layer near the
surface and within the elevated layer where only removal processes influence the $NO_2$
concentrations. As a result, the temporal variation of the concentrations in both layers
was in part influenced by the same processes, differences were not large (Fig. 3a) and
the relationship between the total VCD and NS concentrations was linear with a small
slope (174.24) and well-correlated (R=0.94) (Fig. 3b). In the afternoon, such processes
resulted in the increase of $NO_2$ concentrations, but when the two layers connected and
the wind speed increased somewhat (Fig 2c, d) $NO_2$ was mixed throughout the whole
boundary layer up to about 1000 m. Hence the usual afternoon increase of $NO_2$
concentrations near the surface (Liu et al., 2024) was offset by upward transport,
distributing the $NO_2$ across the whole boundary layer and thus enhancing the total VCD.
This well illustrated by the time series of both NC and total VCD between 13:00 and
15:00 (see Fig. 3a). As a result, the relationship between total VCD and NS changed
substantially after 13:00 (Fig. 3b).

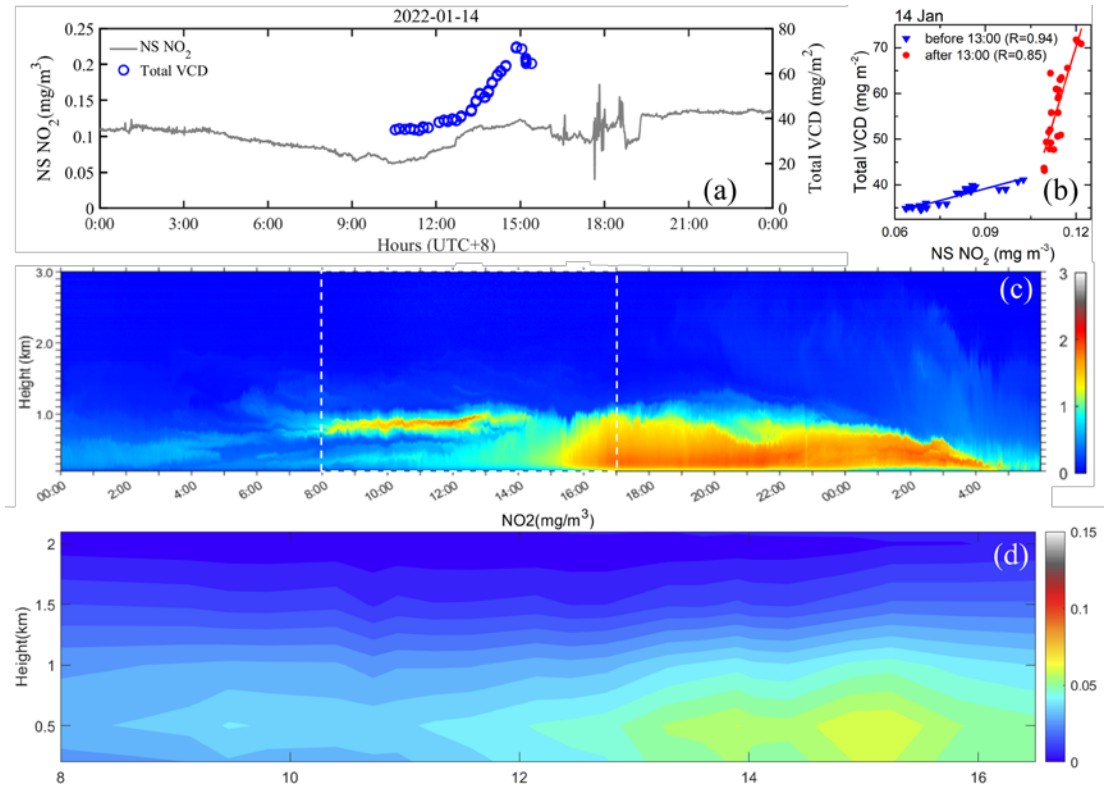

**Figure 3**. (a) Time series of NS $NO_2$ (grey line) and total VCD of $NO_2$ (blue circles) at the Beijing RADI site (40.004°N, 116.379°E) on Jan 14, 2022; (b) scatterplots of total VCD and NS concentrations of $NO_2$ and fits to these data during the morning (before 13:00) and during the afternoon (after 13:00), showing different relationships as discussed in the text; (c) time series of vertical profiles of range-corrected lidar signal at 1064 nm. Note that the lowest height in Fig. 3c is 100 m; (d) time series of $NO_2$ vertical profiles derived from Pandora sky radiance measurements. Note that the Pandora profiles are constructed from layer-averaged volume mixing ratios interpolated to 6 standard levels and the lowest level is 0.2 km.

The effect of transport on the $NO_2$ concentrations at the Beijing-RADI site on 14 January 2022 was analyzed using the data presented in Fig. 4: the spatial distribution of tropospheric $NO_2$ columns derived from TROPOMI data (overpass time 13:30), the synoptic weather map at 00 UTC and 24-hour backward trajectories for arrival at the Beijing-RADI site at altitudes of 300 m, 500 m and 1000 m, at 10:00, 13:00 and 16:00 LT. The TROPOMI data show the relatively high tropospheric $NO_2$ concentrations over the study area, in particular over an elongated area stretching from the SW to the NE

over Hebei Province, including Beijing (compare with Fig. 1), and from Beijing eastward. This area is bounded by the Taihang mountains in the west and by the Yan mountains in the north, blocking transport of pollutants. The weather map in Fig. 4b shows the pressure distribution and location of low pressure areas resulting in wind from the SW, i.e. along the direction of the elongated area with elevated $NO_2$ concentrations (see Fig. 4a). This is confirmed by the air mass trajectories in Fig. 4c, all showing overall transport from the SW. However, the trajectories arriving at 10:00 LT show that during the last 8h, the air mass arriving at 300 m came from the NE at low wind speed and the airmass arriving at 500m came from the NW at even lower windspeed. The airmass arriving at 1000 m came from the SW during the last 14 h before arrival and from the NW during the earlier 12 hours. The airmasses arriving at 13:00 show similar trajectories. These trajectories are consistent with the lidar observation of disconnected layers, with different air mass trajectories during the last hours before arriving at the Beijing-RADI site and thus possibly different composition. The air mass arriving at 1000 m had been at high elevations during its entire 24 h trajectory and originated from higher than 1500m, but those arriving at 300 and 500 m originated from the surface at different locations separated by tens of km and may thus have been influenced by different sources.

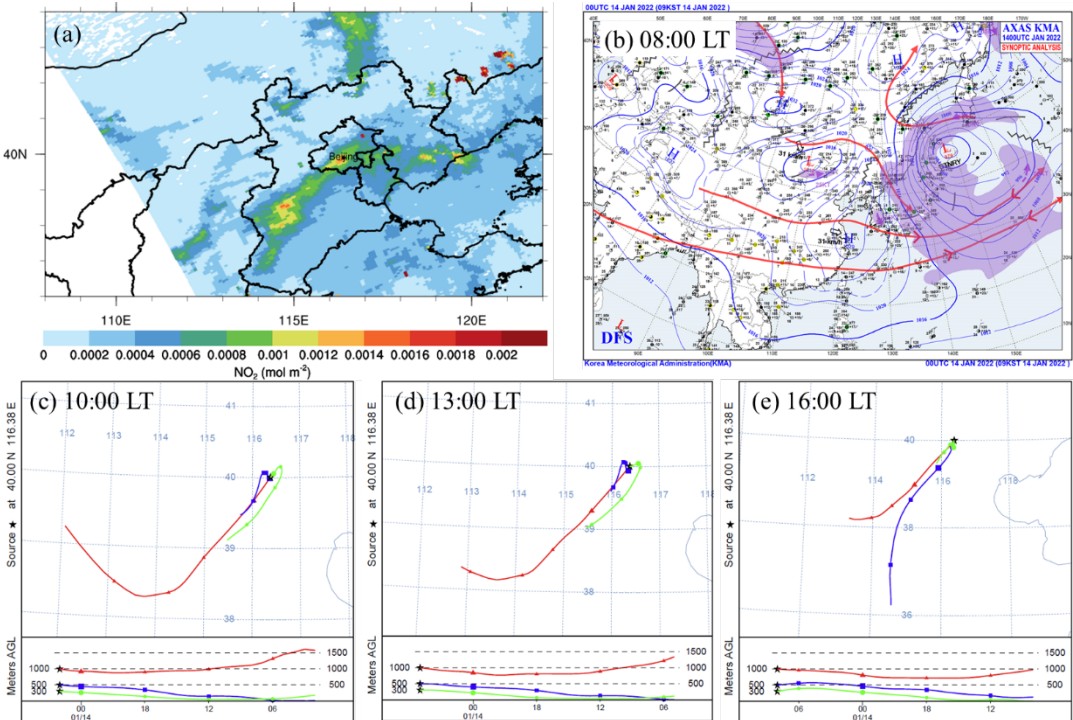

**Figure 4.** (a) Spatial distribution of tropospheric NO$_2$ in the study area derived from TROPOMI data on 14 January 2022; (b) Synoptic weather map at 00:00 UTC (08:00 LT); (c-d) 24-hour backward air mass trajectories arriving at the Beijing-RADI site at 10:00 (c), 13:00 (d) and16:00 LT (e), at heights of 300, 100 and 1000 m, calculated using the HYSPLIT model with 6h time steps (00, 06, 12 and 18) and a shorter time step to the arrival time.

**3.2.2 Case 2: Multi-layer structure on 18 January, 2022**

Time series of the total VCD and NS concentrations of NO$_2$ on 18 January 2022 are shown in Fig. 5a, together with a scatterplot between the total VCD and NS concentrations in Fig. 5b and 3-D plots of the vertical variation of lidar backscatter coefficients in Fig. 5c and Pandora-derived NO$_2$ concentrations in Fig. 5d. The NS NO$_2$ time series in Fig. 5a show that the concentrations were higher than on 14 January, but their variation was initially similar, with a decrease to a minimum around 11:30 (later than on the 14[th]) followed by an increase. However, on 18 January, the increase continued non-linearly until about 14:30 when the concentrations plateaued at a value of about 0.12 mg m$^{-3}$ during about 1 hour and then increased further, likely due to increased emissions during rush hour and decreasing photochemical sink when the solar

radiation intensity decreased in the afternoon. After 18:00 the concentrations plateaued
at 0.13-0.14 mg m$^{-3}$, varied between about 20:00 and 21:00 with values up to about 0.16
mg m$^{-3}$ around 21:00 and decreased somewhat toward the end of the day.

593         The total VCD of $NO_2$ decreased faster than on 14 January, from the initial 54 mg

m$^{-2}$ around 08:30 to the minimum of 36 mg m$^{-2}$ around 12:30, with a smaller decrease
after 11:00. Hence, similar to the situation on 14 January, the total VCD of $NO_2$ initially
decreased while also the NS $NO_2$ concentrations decreased, but in contrast to the 14[th],
after 11:30 the total VCD of $NO_2$ concentrations continued to decrease while the NS
$NO_2$ concentrations increased. As a result, there was no clear correlation between total
VCD and NS concentrations of $NO_2$ before 13:00, as on 14 January. These morning
data could be separated into two groups, before 11:30, when there was no total/NS
relation, and after 11:30 where the data in Fig. 5b suggest a non-linear relation. Hence,
in this situation, it may be difficult to determine NS $NO_2$ concentrations from satellite
data. After 13:00, the total VCD of $NO_2$ increased from about 37 mg m$^{-2}$ to almost 60
mg m$^{-2}$ at 16:30, with a plateau around 15:00. The scatterplot in Fig. 5b shows a good
correlation between TS and NS $NO_2$ concentrations.

606         The lidar data in Fig. 5c, with lower intensity than on 14 January, indicate smaller

aerosol concentrations on 18 January than on 14 January, consistent with the smaller
$PM_{2.5}$ concentrations (in Fig. 2b). The lidar data show the occurrence of multiple layers
during the night and morning, with sharp boundaries indicating that aerosol particles
are trapped in rather shallow layers with little or no exchange between these layers.
After 10:30, the boundaries between layers become less sharp indicating the onset of
vertical transport, although the very shallow clear layer (dark blue) between 500 and
600 m indicates a clear separation between the lower and upper layers, prohibiting
vertical transport. Around 13:00 this shallow layer disappeared and after 15:00 the
atmospheric boundary layer appeared well-mixed up to the top at about 800m.

616         The time series of the $NO_2$ vertical distributions in Fig. 5d shows that $NO_2$

concentrations were lower on the 18[th] than on the 14[th], and concentrated below 1000 m.

The observation of lower concentrations on the $18^{th}$ and the $14^{th}$ seem to be in contrast with the higher NS concentrations on the $18^{th}$ than on the $14^{th}$ mentioned above. However, the Pandora profiles are constructed from layer-averaged volume mixing ratios interpolated to 6 standard levels and the lowest level is 0.2 km. Hence, in view of the layered structure on the $18^{th}$, the higher NS concentrations may be disconnected from the lowest layer set at 0.2 km. It is further noted that the Pandora vertical distributions show lower $NO_2$ concentrations on the $14^{th}$ than on the $18^{th}$ in the morning, whereas they are higher in the afternoon of the $14^{th}$. The $NO_2$ concentrations and their vertical distributions varied between 08:00 and 12:30, with an initial increase between 08:00 and 10:00 with a broad elevated maximum centered around 500 m and another small maximum around 11:00. Apart from these, the $NO_2$ concentrations were rather homogeneously distributed up to the top of the atmospheric boundary layer at about 800 m as also indicated by the lidar data. After about 12:00 the $NO_2$ concentrations increased with a significant enhancement after 14:00.

The evolution of the atmospheric boundary layer, as shown in detail by the lidar data, and the variation of the $NO_2$ concentration profiles provide a plausible explanation for the evolution of the total VCD and NS concentrations of $NO_2$ and their ratios, with changes around 11:30 and 13:00. The plateau in the NS $NO_2$ concentrations may be indicative of the dilution near the surface due to upward transport and vertical mixing, at the same time increasing the TS $NO_2$ concentrations.

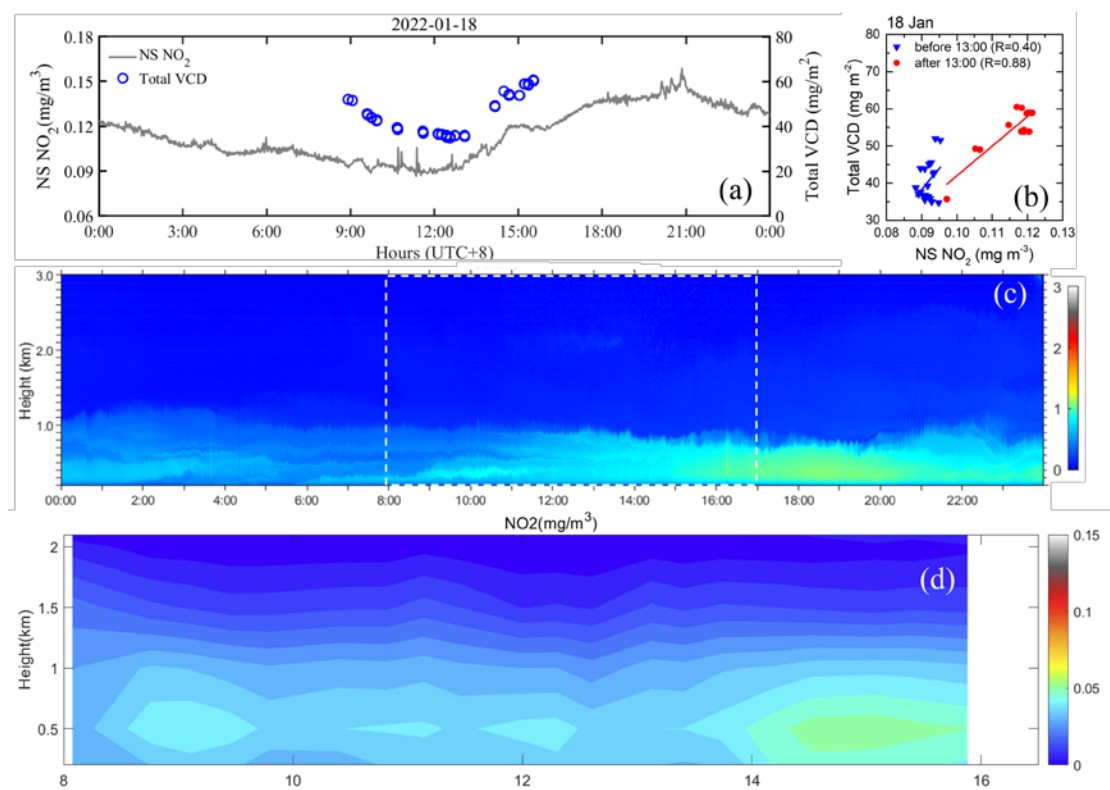

**Figure 5**. Same as Fig. 3 but for 18 January, 2022.

Fig. 6 shows the large scale situation for the study area on 18 January. The TROPOMI data in Fig. 6a show the spatial distribution of the tropospheric $NO_2$ VCDs which are highest to the SE of Beijing, in Hebei/Tianjin and over the Yellow Sea. Over Beijing, the tropospheric $NO_2$ VCDs, as indicated by TROPOMI, are substantially lower than in case 1. This can be explained by the transport from clean areas to the W and WNW of Beijing, as indicated by the air mass trajectories arriving in Beijing at 300m, 500 m and 1000m, at 10:00, 13:00 and 16:00 LT (Fig. 6c). The trajectories of the air masses arriving at 10:00 LT show a clear difference between the lower and higher layers visible in the lidar data: where the air arriving at 1000 m originated from the WNW and had traveled during the last 24 h over clean areas (Fig. 6a) over a distance of 1000 km (10°), between heights of 1000-1500 m, the lower air mass was influenced by local air from SSW (at 300 m) and SW (500m) that had traveled during the last 24 h near the surface at heights up to 500 m over moderately polluted areas. Hence the lidar data show higher aerosol content in the lower layer (<500 m) than in the layer above (>600m) and both disconnected layers are from different origin.

This situation changed as indicated by the air mass trajectory arriving at 12:00 LT. The air mass arriving at 300 m had the same characteristics as at 10:00 LT, had traveled an even shorter distance and the layer adjacent to the surface was more stagnant. However, the air mass arriving at 500 m now came from the west, had traveled over clean areas (Fig 6a) at heights between 500 and 1000 m and thus was distinctly different from the layer below. The air mass arriving at 1000 m originated from a bit further north and further away (in Inner Mongolia) than at 10:00 LT, and had travelled at heights between 750 and 1000 m. Hence all three air masses suggest that the layers originated from different regions and likely had different composition.

The trajectories of the air masses arriving at 16:00 LT indicate that the situation had changed, i.e. the pollution episode was finished and pollution was replaced with cleaner air transported from the W to WNW over distances of hundreds of km and originating from elevations of 500-1000 m for air masses arriving at 300m and 500m, whereas the air mass arriving at 1000 m had actually followed a lower trajectory.

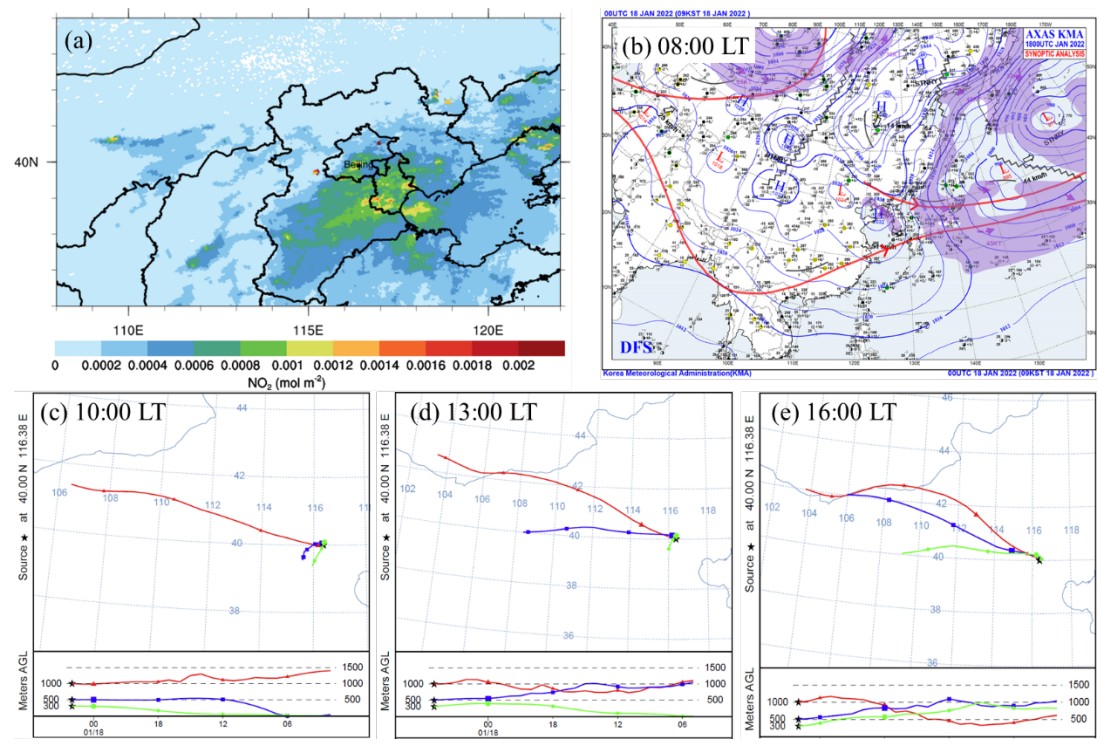

**Figure 6**. Same as Fig. 4, but for 18 January, 2022.

A comparative analysis of the two cases reveals distinct characteristics. Case 1, on

January 14th, is characterized by a belt-shaped pollution event. The air mass arriving at 1000 m was primarily transported from Shanxi to Beijing arriving from the SW during the last 15 hours (Fig. 4c), while at lower levels the air masses travelled at low altitudes (from near the surface to 500 m) through the polluted area in Hebei (compare Figs. 4 a and c). In contrast, Case 2, on January 18th, was a large-scale pollution event covering Shandong, Hebei, and Henan and the southern part of Beijing (Fig. 6a), but the air mass arriving at 1000 m had been transported form the WNW over a large distance over clean areas. However, at lower levels, the air was stagnant (wind speed was low) in the morning, air masses arriving at 300 and 500 m at 10:00 LT had travelled over very short distances during the last 24 hours and were thus only influenced by local pollution. In Case 1, elevated pollutant concentrations were recorded in the 800-1000 m altitude layer in the morning, attributable to an air mass originating from Shanxi Province (with no significant pollution observed there) that was transported over the polluted area in Hebei at about 1000 m (Fig 4a and 4c), and then the upper-level air mass likely carried pollutant residuals which had been uplifted through vertical mixing processes over the polluted area in Hebei. These pollutants were subsequently advected to Beijing, where their presence was detected by lidar observations. Conversely, in Case 2, pollutants were predominantly transported from the plains, leading to a significant accumulation of pollutants in the near-surface layer. After 13:00, in both cases the distinction between the pollutant layers disappeared when the boundary layers developed under the influence of surface heating and increasing wind speed (Fig. 2), thus creating boundary layer turbulence and mixing of $NO_2$, aerosols and other constituents. Both the total VCD and NS concentrations of $NO_2$ increased, with that of total VCD of $NO_2$ being more significant. This further suggests that it is more difficult to obtain NS $NO_2$ concentrations using total VCD of $NO_2$ concentrations during the morning hours. However, utilizing the types of pollution spatial distribution and transport patterns can be helpful in indicating NS $NO_2$ concentrations.

**3.3 Ratio of total VCD vs NS NO$_2$**

The two pollution cases discussed above show that the ratio between total VCD

and NS concentrations of $NO_2$ in the morning (before 13:00) is different from that in the afternoon (after 13:00) during the study period in the winter in Beijing. In order to better understand the relationship between total VCD and NS concentrations of $NO_2$, we calculated their ratio for each day, while we also differentiated between the morning and afternoon using 13:00 as the split time. The Ratio, defined as the ratio of total to NS $NO_2$ concentrations, serves a dual purpose: it not only quantifies the changes between total VCD and NS concentrations of $NO_2$ when the correlation is low but also reflects the degree of dispersion between the two measurements. A more variable Ratio indicates higher dispersion and poorer correlation, providing a straightforward yet effective way to assess the reliability of using total VCD of $NO_2$ to predict NS $NO_2$ concentrations.

The results are presented as violin plots in Fig. 7, for each of the 12 days for which data are available. The data in fig. 7 show that the mean and median values of the $NO_2$ ratio during Period I (10-18 January were substantially higher than those during Period III (25-30 January), with the exception of 25 and 29 January. During these two days, at the beginning and end of Period III, signify the transition from polluted to clean days (see fig. 2b). On most days the ratio was smaller in the morning than in the afternoon. The difference between the morning and afternoon ratios was small during the two days ($14^{th}$ and $18^{th}$) with accumulated pollution, while during the four days when wind speed increased, on 13, 15, and 17, the differences were relatively large, with the largest difference of 192 m on the $13^{th}$. During Period III the difference between the morning and afternoon ratio was basically smaller than 50 m, with a gap of more than 100 m only on the $29^{th}$. There were no valid observations of total VCD of $NO_2$ during Period II, so it is not possible to judge the changes in ratio over multiple consecutive days of pollution. Throughout the observation period, the standard deviations of the ratio were overall larger in the morning than in the afternoon, when the winter boundary layer was well-mixed and the relationship between total VCD and NS concentrations of $NO_2$ was relatively stable. However, in the morning, when the boundary layer was developing, the day-to-day variations in the standard deviation imply relatively large changes in the

ratio. The box plot in fig. 7b illustrates the difference between the morning and
afternoon ratios. The mean values are lower in the morning (364m) than in the afternoon
(428m), and the upper quartile in the afternoon are closer to their median value,
suggesting that the ratio is more stable in the afternoon when it is well mixed vertically.
However, although the ratio is quite stable in the afternoon when the boundary layer is
generally well mixed, there are still unpredictable extreme values (e.g. 771 m).
Generally, the temporal stability of the Ratio is important. The Ratio is overall less
variable after 13:00, suggesting that polar-orbiting satellites can be used to predict NS
$NO_2$ based on total VCD of $NO_2$ during this period with greater confidence. This
temporal stability is particularly valuable because it offers a feasible approach for air
quality monitoring and forecasting. In contrast, the Ratio is less stable before 13:00,
posing greater challenges for using geostationary satellites for the same prediction task.
It's worth noting that our analysis in winter in Beijing suggests that considering both
the spatial distribution of pollutants and their transport direction has the potential to
enhance the ability of satellites to predict NS $NO_2$ concentrations based on total VCD
of $NO_2$. By incorporating this information into prediction models, the accuracy and
reliability of satellite-based air quality predictions may be improved, particularly in
complex urban environments where pollutant concentrations can vary significantly
over short distances and time periods.

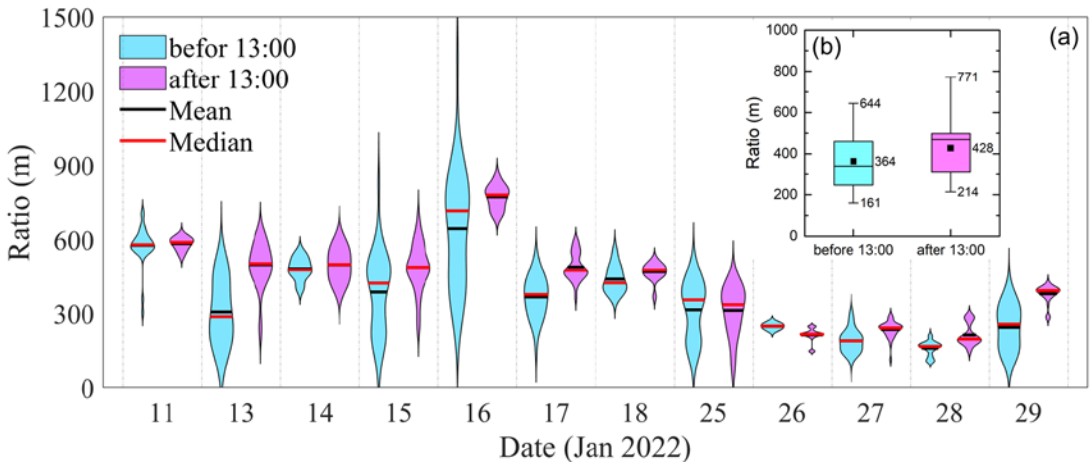

**Figure 7**. (a) Violin plots of Ratio of total VCD and NS concentrations of $NO_2$ for each

day when observations were available in January 2022, where the data were differentiated between morning (before 13:00 LT) and afternoon (after 13:00LT). (b) The box-whisker plot of Ratio averaged over all observations before and after 13:00 in January 2022. The horizontal lines in the boxes and the top and bottom edges represent the mean and upper and lower quartile values of the Ratio, the solid square dots represent the median values, and the bars represent the minimum and maximum values.

## 4 Discussion and Conclusions

Total column and near surface $NO_2$ data observed during the winter field experiment from Jan 10 to Jan 29, 2022, at the Beijing RADI site were analyzed together with lidar, $PM_{2.5}$ and meteorological data, satellite data, weather maps and air mass trajectories. Based on these observations, the experimental period was sub-dived into three periods: intermittent pollution days, persistent pollution days and clean days. The analysis of the total VCD and NS concentrations of $NO_2$ shows substantial differences between the first and third period, while during the second period with persistent pollution no total VCD of $NO_2$ observations were available due to the presence of clouds. During the first period, two episodes with high pollution were identified and analyzed in detail with a focus on the ratio between the variation of the total VCD and NS concentrations of $NO_2$ and their ratio. The relations between the total VCD and NS concentrations of $NO_2$ in the morning and in the afternoon, split at 13:00 LT, appear to be significantly different. These differences have been explained in terms of boundary layer dynamics, using lidar data showing the vertical stratification with disconnected boundary layers at different heights in the morning which connected as the boundary layer developed in the afternoon. In addition, the 4-layer $NO_2$ column concentrations obtained from Pandora show good agreement with the lidar signal in terms of the temporal and vertical variations of the $NO_2$ concentrations, with differences attributed to the different vertical resolutions of the Pandora and lidar observations, as well as physical properties of $NO_2$ and aerosols. From this, together with air mass trajectories, weather maps and TROPOMI satellite observations of the $NO_2$ spatial distribution, a consistent picture was created showing different source regions for disconnected airmasses arriving at

different heights and different times of the day.
Data from the full experimental period, with 12 days for which valid data are available,
were analyzed in detail to obtain more insight into the variation of the ratio between the
total VCD and NS concentrations of $NO_2$. This ratio appears to be overall smaller in the
morning than in the afternoon, with larger standard deviations. In addition, the ratios
and their standard deviations were overall larger during the more polluted episode I
than during the relatively clean period III.
Day-time continuous remote sensing observations of Pandora were used in this study
and the results confirm its possible importance in understanding changes in the
distribution of $NO_2$ in the vertical direction. The $NO_2$ vertical distribution has been
analyzed using less than 3 weeks of observation data, which has some limitations, but
the research idea is worthy of reference and promotion. In the future, the
implementation of larger-scale experiments in different typical regions and seasons will
help to provide further understanding of the ideas presented in this study and improve
the shortcomings. Moreover, we will broaden the scope of experimental areas and field
sites to complement research on the various pollutant emission and transport
characteristics. Furthermore, observations over a longer period will allow us to capture
more representative cases, thereby enhancing the reliability of our findings.
The overall conclusion from this study during a relatively short period of almost 3
weeks in the winter in Beijing is that the variation between the total column and near
surface $NO_2$ concentrations varies with the concentration level and the time of day. In
the afternoon the boundary layer is well developed and satellite observations are
sensitive to the NS concentrations, whereas in the morning this depends on
meteorological conditions. Hence, satellites with an afternoon overpass are capable to
measure total VCD of $NO_2$ which is representative for NS concentrations, whereas
observations earlier in the day may not be. This could possibly affect the interpretation
of diurnal variations derived from observations using geostationary satellite.


*Data availability*. Data will be made available on request.

*Author contributions*. YZ and YW conceived and designed the study. OL processed the Pandora data. YC collected and processed the meteorological data. YL processed the Lidar data. YZ and GL prepared the paper with contributions from all coauthors. Y-XZ and ZL provide project funding.

*Competing interests.* The authors declare that they have no conflict of interest.

*Acknowledgements.* We thank the HYSPLIT development and maintenance team, the Beijing RADI site maintainers, and the TROPOMI product development and maintenance team for their support. The PGN is a bilateral project supported with funding from NASA and ESA.

*Financial support.* This work was supported by the National Key Research and Development Program (2022YFE0209500), the National Natural Science Foundation of China (42101365) and the Chinese Academy of Sciences President's International Fellowship Initiative. Grant No. 2025PVA0014.

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
