# Peer review of "Relation between total-column and near-surface NO$_2$ based on in-situ and PANDORA ground-based remote sensing observations"

_EGUsphere, 2025_

## Author Comment (AC1)

**Response to comments by Reviewer #1**

We thank Reviewer #1 for the detailed review of our manuscript. The suggestions and comments helped us to further improve the revised version. Below we copy each comment and respond to each of them in blue fonts.

This manuscript evaluated the relationship between total-column and near-surface NO2 in winter in Beijing in using in-situ and PANDORA ground-based remote sensing observations. Detailed vertical distribution was obtained from Lidar observations. Then the possible influence of the atmospheric boundary layer evolution on the near-surface pollutants was discussed. Also, the coupling mechanism between the transport of pollutants and the complex vertical distribution of the atmospheric boundary layer was analyzed using back trajectories and weather maps. The manuscript is well-written and fits the scope of the journal. It provides an interesting case study of the vertical distribution changes of pollutants in the boundary layer based on remote sensing technology. However, there are a few aspects that need to be addressed to further improve the quality of the manuscript.

1. The manuscript highlighted the use of PANDORA for stratification inversion. This is an interesting point, but a detailed explanation of its credibility as well as its role in practical applications are lacking. Please further elaborate on the advantages of PANDORA technology to state its importance and accuracy to readers.

**Response:** Thank you for your comment. Here, based on the technical methodology outlined in the ATBD, we briefly introduce the process of the gas profiling retrieval algorithm using PANDORA as follows:

The first step in determining the partial column amounts of $NO_2$ in different layers is to estimate the effective height (center) corresponding to a given Pointing Zenith Angle ($PZA_i$), which is based on the air-gas slant columns using geometrical relationship and assuming single scattering conditions only:

$$h_{CENi} = \frac{\Delta qs_{i,AIR} + q_{CLIM,AIR}}{n_{SURF,AIR}} \cdot \frac{cos(PZA_i)}{2}$$

Where, $\Delta qs_{i,AIR}$ is the absolute slant column amount of the air-gas at $PZA_i$. $q_{CLIM,AIR}$ is absolute vertical column amount of the air-gas from a climatology. $n_{SURF, AIR}$ is the climatological surface concentration of the air-gas. The measured differential slant columns at $n_{PZA}$ angles ($i=1$ to $n_{PZA}$) give us $n_{PZA-1}$ atmospheric layers. The top height of layer i is given by:

$$h_{TOPi} = 0.5 \cdot (h_{CENi+1} + h_{CENi})$$

The lowest layer extends from the surface to $h_{TOP1}$, where the corresponding angles are $PZA_{MAX}$ and $PZA_{MAXm1}$. The next step is to calculate differential air mass factors for $NO_2$ and the air-gas for each layer, $\Delta m_i$ and $\Delta m_{i,AIR}$ respectively:

$$\Delta m_i = \frac{qs_i - qs_{i+1}}{q_T}$$

$$\Delta m_{i,AIR} = \frac{qs_{i,AIR} - qs_{i+1,AIR}}{q_{CLIM,AIR}}$$

Where, $qs_i$ and $qs_{i,AIR}$ is the absolute slant column amount of the NO2 and air-gas at $PZA_i$. The profile shape $p_s$ of the partial columns is determined as a variation of the air-gas shape:

$$ps_i = \frac{0.5 + \Delta m_i - \Delta m_{i,AIR}}{1 + \Delta m_{i,AIR}} \cdot q_T$$

Where, $q_T$ is the tropospheric column. For the normalized profile shape psn, the profile shape is divided by its largest element, hence all $psn_i$ are $\leqslant 1$. The average number density of the $NO_2$ in layer i, $n_i$, is then calculated from the following empirical equation:

$$n_i = \frac{2 \cdot (\Delta qs_i - \Delta qs_{i+1}) + 0.5 \cdot q_T \cdot (m_{i,RAYL} - 3 \cdot \Delta m_{i,AIR})}{\Delta qs_{i,AIR} - \Delta qs_{i+1,AIR} + q_{CLIM,AIR}} \cdot n_{SURF,AIR} \cdot psn_i$$

$m_{i,RAYL}$ is the Rayleigh air mass factor for the corresponding PZA. It is calculated from Radiative Transfer Calculations. Further correction due to profile differences are addressed by the correction factor $cf_i$:

$$cf_i = psn_i \cdot \frac{n_i}{Max(n_i)}$$

Max() refers to the maximum over all i. For those layers with negative values of $cf_i$ the $n_i$ are corrected in the following way:

$$n_i = n_i \cdot (1 + cf_i)$$

Negative values of $n_i$ are set to 0. For the partial column amounts $\Delta q_i$, the concentrations $n_i$ are multiplied with the layer thickness:

$$\Delta q_i = n_i \cdot (h_{TOPi} - h_{TOPi-1})$$

For i=1, the lowest layer, $h_{TOPi-1}$ equals 0. The last step is to normalize the $\Delta q_i$ with the tropospheric column:

$$\Delta q_i = \Delta q_i \cdot \frac{Sum(\Delta q_i)}{q_T}$$

Sum() refers to the sum over all i.

Regarding the accuracy of the $NO_2$ TranProf product at four levels (0-4 km), both official PANDORA documents, "Fiducial Reference Measurements for Air Quality" and "Pandonia Global Network Data Products Readme Document," clarify that there is currently no accuracy assessment for this product. When using it, we followed the accuracy screening criteria for L1 products and retained the sections with high accuracy in the radiometric observations.

The $NO_2$ profile inversion is explained by modifying the text at the end of Sect. 2.2.1 to:

Line 225-241: "Diffuse (scattered) radiation is measured at 5 pointing zenith angles (PZAs) in sky

mode which, together with the direct sun measurement, provides information on the tropospheric VCD and on the surface concentrations. The PZAs are 0º, 60º, 75º, 88º and a maximum angle taken as 89º. The measurements are taken in a V shape (all angles are measured twice around a central angle) as described in Cede (2024). Four partial columns of $NO_2$ concentrations are provided by the PANDORA inversion. The first step is the estimation of the effective height corresponding to a given PZA, and then calculate differential air mass factors for the $NO_2$ and the air-gas for each layer. The profile shape of the partial columns is determined as a variation of the air-gas shape. The average number density of the $NO_2$ in each layer is then calculated. The partial column amounts can be obtained from the concentrations multiplied with the layer width as described in the Manual for Blick Software Suite (Cede, 2024), Section 6.7. The $NO_2$ of the partial column can be obtained from the uvh3 product which was downloaded from the PGN website (https://pandonia-global-network.org, last accessed: 22 Jan 2025). We converted these partial column concentrations into layer-averaged volume mixing ratios and interpolated them to 6 standard levels (0.2, 0.5, 1.0, 1.5, 2.0, 2.5 km) for visualization."

2. The authors used total concentrations when analyzing NO2 obtained from ground-based remote sensing, but compared the tropospheric concentrations of NO2 with that from satellite sensor TROPOMI. This needs to be clarified.

**Response:** Thank you for your comment. Firstly, we used TROPOMI satellite observations of tropospheric $NO_2$ to confirm that the pollutants originated from regional pollution transport processes, rather than to strictly compare satellite data with ground-based measurements. Secondly, we reviewed the TROPOMI satellite products and found that only tropospheric products were available during the time period of our study. Finally, in our previous research (Liu et al, 2024), we have compared total vertical column densities (VCDs) and tropospheric VCDs of $NO_2$ from TROPOMI with PANDORA using 2022 data, and obtained good validation results. For clearer description in the manuscript, we have added an explanation as follows:

Line 269-273: "In addition, only tropospheric vertical column densities (VCDs) for $NO_2$ were available during the time period of this study. The TROPOMI tropospheric VCDs for $NO_2$ are only used as a qualitative reference for upwind concentrations for the evaluation of effects of long-range transport using air mass backward trajectories and not for quantitative analysis."

3. The observation was divided into three periods, in which PM2.5 continued to rise during Period II and suddenly dropped dramatically on the 25th. In the case study, the authors only chose the two cases of Period I. The cases during Period II were not analyzed. Please explain this in detail.

**Response:** Thank you for your comment. There was indeed a persistent heavy pollution process during Period II. However, due to the occurrence of clouds during that period (as mentioned in Sect. 3.1 , line 340 "total VCD of $NO_2$ observations are not available due to the presence of clouds"), we are unable to obtain available observations in cloudy or severely hazy weather conditions. This is a limitation of the use of optical remote sensing and we were unable to select case studies from Period II. We have stated this reason in the manuscript as follows:

Line 413-418: "Our study primarily focuses on the correlation between total VCD and NS concentrations of $NO_2$. Total VCD of $NO_2$ are provided by the passive remote sensing instrument PANDORA, which does not provide reliable observations on cloudy days, as mentioned above. This is the reason why we only selected cases during Period I and did not obtain cases during Period II,

when pollution was more severe."

4. In the case studies of the 14th and 18th, the PANDORA remote sensing only had observations during the daytime. This makes sense. However, there was a difference in the vertical distribution of NO2 from PANDORA versus Lidar. I can understand that the Lidar signal comes more from the scattering of aerosol particulate matter, but the vertical decoupling of NO2 does not appear in the PANDORA observations. Although the authors have some explanation for this, I think it needs further clarification of these differences.

**Response:** Thank you for your constructive comment. PANDORA retrieval provides $NO_2$ column concentrations by dividing the atmospheric column into 4 layers, for which the reported height represents the top of each layer. In Figures 3d and 5d of the manuscript, we converted these column concentrations into layer-averaged volume mixing ratios (from mg m$^{-2}$ to mg m$^{-3}$ units) and interpolated them to 6 standard levels (0.2, 0.5, 1.0, 1.5, 2.0, 2.5 km) for visualization (for more detail see our response to your 1$^{st}$ comment). To clarify the vertical resolution limitations of PANDORA, we now present the original 4-layer data in Figure R1. The results show: When $NO_2$ column concentrations are elevated, the retrieved layer top heights decrease significantly, with high-value concentrations primarily distributed between 0.2-1.0 km and increasing with time during a day. Lidar observations reveal complex sub-kilometer stratification within the 1 km boundary layer, with a resolution of 7.5 m, where PANDORA's coarse 4-layer resolution limits its ability to resolve fine-scale $NO_2$ gradients. This was mentioned in the original manuscript on lines 777-778: "with differences attributed to the vertical resolution of the Pandora and lidar observations". For further clarification, we have added the following explanation to the manuscript:

Lines 512-517: " These differences stem from the difference in vertical resolution between the PANDORA and the lidar: the total VCD from PANDORA is divided into two layers (approximately 300-800 m and 800-1700 m) within the detailed stratified height range (300-1000 m) observed by lidar. Consequently, the fine stratified structure within 1000 m cannot be identified with the available PANDORA data."

[Figure]

Figure R1. NO₂ column concentrations retrieved by PANDORA

5. Based on the multi-temporal backward trajectory and weather maps and in conjunction with the remote sensing imagery from TROPOMI, both cases were in good agreement with the regional transport pattern. In the vertical distribution evolution, it was also consistent with the expectation of winter boundary layer development. Although the authors have explained the boundary layer

development and the conformity of the backward trajectories at different heights, the differences between the two cases were not yet clearly summarized. Despite the presence of multiple layers in the boundary layer in both mornings, there is a correlation between the distribution of pollution concentrations in the layers and the regional pollution intensity and transport direction, which is presented in both cases. It would be great if the authors could verify whether this is the case and summarize the differences, which I believe would be interesting.

**Response:** Thank you for your comment. We did not clearly articulate the basis for selecting the cases or their characteristics. In fact, we chose typical cases based on the pollutant transport paths in Beijing. Following Yin et al. (2025), SW (Shanxi-Beijing) and SE (Shandong-Hebei-Beijing) are two common pollutant transport pathways affecting Beijing.

[Figure]

Fig R1. The wind field and pollutant absolute contribution from different regions under SW and SE patterns. (Yin et al., 2025)

Case 1 (January 14th) represents a belt-shaped pollution event mainly transported from Shanxi to Beijing, as indicated by the elongated SE-NW pattern (Fig. 4a). The 24-hour air mass trajectories (Fig. 4c) confirm the overall transport from the SW, with the lower two trajectories close to the surface during the first 12 hours. These two lower trajectories (arriving at 300 and 500 m), clearly indicate the stagnating air mass, but coming from different direction during the last few 6-hour steps: the air mass trajectory arriving at 300 m travelled from the NE during the last 12 hours while the air mass arriving at 500 m travelled even shorter distances during the last 18 hours and arrived from the NW. These trajectories clearly show the separation of air masses at different heights, passing over different areas and thus with different influences as regards picking up pollutants from different sources. And possibly these pollutants are influenced by different processes since wind speeds were different, and other influencing meteorological factors may have been different as well.

Case 2 (January 18th) features a large-scale pollution event extending from the southern part of Beijing into Shandong, Hebei and Henan) (Fig. 6a). The air mass trajectories (Fig. 6c) show the overall transport from the WNW over large distances (10º) for the air mass arriving at 1000 m, but with changing patterns for the airmasses arriving at 300 and 500 m. Initially, at 10:00, these air mass show the stagnating air, with very low windspeed from the S (300 m) to the SW (500 m). At 13:00, the 500 m air mass trajectory originated from 1000 m (as opposed to the air mass arriving at 10:00 that originated near the surface and travelled at 500 m) and had travelled about 8º from the west during 18 hours when it stagnated during the last 6 hours near the arrival location. The air mass arriving at 300 m had travelled a very short distance from the SSW and stagnated near the arrival point. The air masses arriving at 16:00 all had travelled longer distances (8º or more) and arrived from the W during the last 6 hours and earlier from the W (300 m), WNW (500 m) and NW (1000 m). This clearly marks the end of the pollution episode with increasing wind speeds, transporting cleaner air from the W and NW and ventilating the pollution out. Also in this case the air mass

trajectories indicate very different origins of the air mass at different heights, and thus different pollution sources.

In Case 1, higher pollutant concentrations were observed in the 800-1000 m layer (Fig 3a) because the air mass primarily transported from Shanxi to Beijing along the SW direction, while the pollutants were carried by the air mass as it passed through Shijiazhuang City in Hebei Province. In Case 2, pollutants were mainly transported from the plains, resulting in higher pollutant concentrations in the near-surface layer. These characteristics align well with the spatial distribution and vertical stratification of pollution, and both cases are highly representative. We have summarized the characteristics of the two cases as follows:

Line 672-699: "A comparative analysis of the two cases reveals distinct characteristics. Case 1, on January 14th, is characterized by a belt-shaped pollution event. The air mass arriving at 1000 m was primarily transported from Shanxi to Beijing arriving from the SW during the last 15 hours (Fig. 4c), while at lower levels the air masses travelled at low altitudes (from near the surface to 500 m) through the polluted area in Hebei (compare Figs. 4a and 4c). In contrast, Case 2, on January 18th, was a large-scale pollution event covering Shandong, Hebei, and Henan and the southern part of Beijing (Fig. 6a), but the air mass arriving at 1000 m had been transported form the WNW over a large distance over clean areas. However, at lower levels, the air was stagnant (wind speed was low) in the morning, air masses arriving at 300 and 500 m at 10:00 LT had travelled over very short distances during the last 24 hours and were thus only influenced by local pollution. In Case 1, elevated pollutant concentrations were recorded in the 800-1000 m altitude layer in the morning, attributable to an air mass originating from Shanxi Province (with no significant pollution observed there) that was transported over the polluted area in Hebei at about 1000 m (Fig 4a and 4c), and then the upper-level air mass likely carried pollutant residuals which had been uplifted through vertical mixing processes over the polluted area in Hebei. These pollutants were subsequently advected to Beijing, where their presence was detected by lidar observations. Conversely, in Case 2, pollutants were predominantly transported from the plains, leading to a significant accumulation of pollutants in the near-surface layer. After 13:00, in both cases the distinction between the pollutant layers disappeared when the boundary layers developed under the influence of surface heating and increasing wind speed (Fig. 2), thus creating boundary layer turbulence and mixing of $NO_2$, aerosols and other constituents. Both the total VCD and NS concentrations of $NO_2$ increased, with that of total VCD of $NO_2$ being more significant. This further suggests that it is more difficult to obtain NS $NO_2$ concentrations using total VCD of $NO_2$ concentrations during the morning hours. However, utilizing the types of pollution spatial distribution and transport patterns can be helpful in indicating NS $NO_2$ concentrations."

6. I have some confusion about Figure 7. Although it showed regular data, the ratios were not the correlation coefficients or slopes that were used in the case studies. The slopes were fixed in the cases where TC correlates well with NS NO2 concentrations, whereas the ratios were not slopes, which were highly variable. I think the authors need to give a more detailed explanation of how they are related and different.

**Response:** Thanks for your comment. We apologize for this confusion and try to explain better why we introduced Ratio. The analysis of the two cases in Section 3.2 shows the good correlation between total VCD and NS $NO_2$ concentrations in the afternoon, and for Case 1 also in the morning, but not for Case 2. This prompted us to look also at other days to determine whether and when a good correlation occurs and whether a linear relationship between total VCD and NS $NO_2$

concentrations, characterized by $y = ax + b$, can be established. This exercise showed that for a larger number of days the correlation is less good than in cases 1 and 2, particularly during the morning hours. The low correlation does not quantify the changes between total VCD and NS $NO_2$, prompting us to introduce the Ratio as a means of quantification. To further explore a relation between total VCD and NS concentrations of $NO_2$, we introduce the Ratio between these variables, Ratio = total VCD / NS, and looked at the variation of Ratio. The data shows that the variability of Ratio is generally smaller in the afternoon than in the morning, which indicates the feasibility of using polar-orbiting satellites with an afternoon overpass time to determine NS $NO_2$ based on total VCD of NO2. However, using geostationary satellites for the same prediction based on total VCD observations before 13:00 poses greater challenges. Notably, the analysis of cases 1 and 2 showed that regional pollution transport paths may be helpful for predicting NS concentrations of $NO_2$ in winter in Beijing. In other words, considering both the spatial distribution of pollutants and their transport path ways in predictions has the potential to enhance the ability of satellites to predict NS concentrations $NO_2$ based on observations of total VCD of $NO_2$. The limitation of our study is that it is based on only one month of observations, and during this period, the geostationary satellite GEO-KOMPSAT-2B /GEMS did not release any products, which prevented us from better demonstrating the universality and applicability of our findings. In future research, we will obtain more observations and utilize geostationary satellite observation products to further refine and validate our discoveries.

We further elaborated on the role of Ratio in the MS as follows:

Line 706-712: "The Ratio, defined as the ratio of total to NS $NO_2$ concentrations, serves a dual purpose: it not only quantifies the changes between total VCD and NS concentrations of $NO_2$ when the correlation is low but also reflects the degree of dispersion between the two measurements. A more variable Ratio indicates higher dispersion and poorer correlation, providing a straightforward yet effective way to assess the reliability of using total VCD of $NO_2$ to predict NS $NO_2$ concentrations."

We further summarized the comprehensive point presented by the analysis of Ratio and the two cases as follows:

Line 737-749: "Generally, the temporal stability of the Ratio is important. The Ratio is overall less variable after 13:00, suggesting that polar-orbiting satellites can be used to predict NS $NO_2$ based on total VCD of $NO_2$ during this period with greater confidence. This temporal stability is particularly valuable because it offers a feasible approach for air quality monitoring and forecasting. In contrast, the Ratio is less stable before 13:00, posing greater challenges for using geostationary satellites for the same prediction task. It's worth noting that our analysis in winter in Beijing suggests that considering both the spatial distribution of pollutants and their transport direction has the potential to enhance the ability of satellites to predict NS $NO_2$ concentrations based on total VCD of $NO_2$. By incorporating this information into prediction models, the accuracy and reliability of satellite-based air quality predictions may be improved, particularly in complex urban environments where pollutant concentrations can vary significantly over short distances and time periods."

7. There are also some other minor issues that need further improvement, such as the format of the citation.

Line 167: "… north, east, and west (Figure 1). ", for figures, the manuscript used fig more often, please make sure that they are consistent.

**Response:** We have modified it to "… north and west (Fig. 1). " and checked the whole manuscript.

Line 205: "… precision of 0.01 DU and a nominal accuracy of 0.1 DU Herman et al. (2009)", the citation should be in parentheses as a whole.

**Response:** The manuscript has been revised.

Line 260: "PM2.5 is measured using beta rays generated", "2.5" should be subscripted.

**Response:** The manuscript has been revised.

Line 304: "e.g., Atkinson, 2000; Boersma et al., 2009; Y. Zhang et al., 2016", "Y. Zhang" should be "Zhang". Please double check all the citations so that the format is consistent.

**Response:** The manuscript has been revised.

Line 442: "… with a wind speed of 2 m/s", for units, superscripts are used more often than slashes throughout the manuscript. Please revise it to be consistent.

**Response:** The manuscript has been revised.

---

## Author Comment (AC2)

**Response to comments by Reviewer #2**

We thank Reviewer #2 for the detailed review of our manuscript. The suggestions and comments helped us to further improve the revised version. Below we copy each comment and respond to each of them in blue fonts.

The manuscript "Relation between total-column and near-surface NO2 based on in-situ and PANDORA ground-based remote sensing observations" by Zhang et al. investigates the relationship between total column (TC) and near-surface (NS) NO2 concentrations using field experiments, supported by ancillary data and model analysis. The study highlights the value of Pandora observations in capturing and understanding the dynamic vertical distribution of NO2. Additionally, the use of a backward trajectory model in two case studies effectively illustrates air mass motions at different altitudes, providing meaningful insights into the evolution of TC-NS relationships.

Overall, the study is well-structured and employs a clear methodology. However, concerns regarding its novelty, broader implications, and generalizability may limit its impact. Therefore, major revisions are required to justify its publication in Atmospheric Chemistry and Physics (ACP).

Major comments:

(1) The study is based on a single station (Beijing-RADI) and a short observation period (January 10-29, 2022), which limits the applicability of the results to other regions and seasons. Since the authors emphasize the complexity of the TC/NS NO2 relationship, it raises concerns that this limited dataset may not fully capture its variability. It is suggested that the authors provide a detailed justification for the data selection, explaining why the short-term observations are still appropriate for investigating this relationship. Additionally, the representativeness of the case study should be explicitly discussed, particularly whether the findings are expected to hold across different meteorological conditions and locations. If possible, a comparison of the TC-NS NO2 relationship between the Beijing-RADI site and other sites is recommended to strengthen the generalizability of the conclusions.

**Response:** Thanks for your comment. We agree with your point, yet our considerations are as follows: firstly, Beijing was selected as the study area due to the persistent pollution in the Beijing-Tianjin-Hebei (BTH) region, where polluted days accounted for 36% of the year in 2020, with 3% of these days experiencing heavy pollution (China Ministry of Ecological Environment, 2020). This highlights the importance of research in this area. Secondly, Beijing, as the capital of China, is a megacity with a vast population. Surrounded by mountains to the northwest, polluted weather forms easy during southwest, southeast, or calm wind conditions. Consequently, many studies have focused on Beijing or the urban agglomeration of the BTH region. In recent years, based on research on fine particulate matter, the types of pollutant transport in the Beijing area have been deeply investigated. For instance, Yin et al. (2025) indicated that there are three pollution types for Beijing: SW, SE, and SM. The SW type involves pollutants being transported from Shanxi through Hebei to Beijing, while the SE type primarily involves pollutants being transported from Shandong and the eastern side of Hebei, as well as Tianjin. Other studies (Dong et al., 2020; Chang et al., 2019; Li et al., 2017) have also reached consistent conclusions. Most of their analyses are based on multi-year pollution classification in the Beijing area and are representative, laying a foundation for our

research. Given limited experimental funding and conditions, we selected January, the winter season in Beijing, as the study period based on the patterns summarized in these studies. We conducted comprehensive ground-based measurements using various instruments, complemented by the first Pandora spectrometer in China and satellite observations, developed the study of vertical pollution and regional transport characteristics in Beijing. As previous studies have reported the contributions of different pollution transport types to the Beijing area, our objective is to identify representative cases to illustrate the impact of different transport types on the vertical structure of $NO_2$ and its diurnal evolution patterns. This can provide scientific explanations for the increasing number of satellite remote sensing studies on NS $NO_2$ concentrations and also suggest that future research using geostationary satellites to study NS $NO_2$ concentrations should focus on vertical variations within the daytime boundary layer. We appreciate the comment of expanding observations (time period and stations), but due to observational limitation (Pandora and Lidar are expensive instruments), this is currently not feasible. In the future, we will strive to develop as many comprehensive observation stations as possible and collect more representative cases to further address the scientific questions focused on in this study. In order to further explain our ideas, we have made the following modifications to the MS:

Line 403-410: "Many studies (Yin et al., 2025; Dong et al., 2020; Chang et al., 2019; Li et al., 2017) indicated that the pollution in Beijing mainly originates from three pollution transport patterns: southwest (SW), southeast (SE), and south mixing (SM) patterns. Among them, the most representative are the SW transport path from Shanxi province to Shijiazhuang-Baoding-Beijing and the SE transport path from Shandong province to Cangzhou-Langfang-Tianjin-Beijing. However, few studies have explored the vertical transport of pollution types. Cases 1 and 2 are representative for the widespread SW and SE patterns, respectively."

(2) The study emphasizes the importance of investigating the TC-NS NO2 relationship for satellite-based monitoring of NS NO2 concentration and discusses how the TC/NS ratio changes with time and meteorology. However, the manuscript does not directly analyze the relationship between satellite TC NO2 and NS NO2, instead focusing solely on the Pandora TC NO2 and NS NO2. Given that the authors acknowledge biases between Pandora and satellite TC NO2 in the introduction, it is recommended that additional analyses be included to assess how TC/NS variations impact actual satellite NO2 applications on NO2 monitoring.

**Response:** Thanks for your comment. First, to avoid ambiguity, we refer to TC $NO_2$ as the total vertical column density (VCD) of $NO_2$, and the manuscript has been thoroughly revised accordingly. For polar-orbiting satellite observations, we have compared the total VCD and tropospheric $NO_2$ concentrations obtained by TROPOMI and Pandora from August 2021 to July 2022 in our previous study (Liu et al., 2024) and found a good correlation (Fig. R1), as mentioned in the Introduction (Lines 96-99). Figure R1 demonstrates that the correlation coefficient R between TROPOMI and PANDORA for both the total VCD of $NO_2$ and for the tropospheric VCD of $NO_2$ exceeds 0.9. In comparison, the mean difference (MD) between total VCD of $NO_2$ concentrations from TROPOMI and Pandora is 8.44%, while tropospheric $NO_2$ concentrations exhibit an MD of 16.15%. For absolute differences, the total VCD has a slightly higher value than the tropospheric results, with mean absolute differences (MAD) of 3.14 and 2.71 Pmolec $cm^{-2}$, respectively. For error distribution (Fig. R1b&d), the tropospheric $NO_2$ is closer to a normal distribution, while the mean error of total VCD of $NO_2$ shows a shift of -1.16 Pmolec $cm^{-2}$.

It is noteworthy that during the field experiment period in January 2022, we obtained observations from a Trace Gas Analyzer and a Lidar, but there were no total VCD of NO₂ products available from neither TROPOMI nor GEMS during that same period. This posed a certain obstacle to our research. However, the main objective of this study is to capture the vertical distribution changes of golden cases and understand the vertical distribution differences. Therefore, we believe that the comparison of satellite data can be deferred to the next stage of research, and this stage should focus on analyzing the temporal-spatial complexity of NO₂. Fortunately, we did identify two such cases during the field experiment. Although these cases do not cover all the major pollution transport types (SW, SE, SM) mentioned in previous studies (e.g. Yin et al., 2025) in Beijing, we have explained the temporal evolution of horizontal and vertical pollution within the limited observations. We believe that this is meaningful.

[Figure]

Figure R1. Validation of TROPOMI total and tropospheric NO2 VCDs, resampled to a spatial resolution of 100m×100 m, using Pandora observations as reference data: (a, c) scatterplots of total and tropospheric TROPOMI vs. Pandora data together with statistical metrics; (b, d) histograms of the differences between TROPOMI and Pandora NO2 total / tropospheric VCDs. (Liu et al., 2024, Figure 9)

(3) The manuscript does not clearly establish the novelty of the study compared to prior works. The motivation expressed in lines 145-148 is not sufficiently developed to justify the study's significance. While the paper states that accurate TC/NS NO2 information was not previously available for China, it does not explicitly explain why this makes the study novel or how it differs from previous research on TC/NS relationships. To strengthen the motivation, the authors should clarify what specific gaps in the literature they are addressing and explicitly compare their approach to existing studies.

**Response:** Thanks for your comment. I am very sorry that we didn't clearly articulate the innovation of the research. We believe that the rapid advancement of satellite remote sensing technology currently enables regional detection of atmospheric pollutants. However, satellite remote sensing primarily captures the pollutant content within the atmospheric column, whereas it is the nearsurface pollution that directly impacts human health. Numerous studies (Chang et al., 2025; Wei et al., 2022; Zhang et al., 2022; Dou et al., 2021) have been conducted on converting the pollutant content in the atmospheric column obtained by satellite remote sensing into near-surface concentrations. Yet, a crucial issue arises: the complex evolution of the local boundary layer limits the accuracy of estimating near-surface pollutant concentrations. Therefore, our research aims to utilize ground-based remote sensing observations and near-surface $NO_2$ concentration measurements to investigate vertical decoupling of $NO_2$ within the boundary layer during winter in Beijing and discuss the influence of regional pollutant transport on the vertical distribution of $NO_2$ concentrations. Although there are numerous studies on the vertical distribution of $NO_2$ (Sun et al., 2023; Zhang et al., 2023; Kang et al., 2021) and the contribution of regional transport (Yin et al., 2025; Dong et al., 2020; Chang et al., 2019; Li et al., 2017), respectively, research linking regional pollutant transport to the vertical distribution of $NO_2$ concentrations and clarifying the impact of regional transport on vertical concentration distributions remains scarce. Focusing on this key aspect, we propose possible connections between regional pollutant transport and vertically stratified $NO_2$ concentrations, with the aim of providing valuable information for satellite remote sensing to predict near-surface $NO_2$ concentrations. We have added the following text to Section 1, describing gaps that are addressed in this study and resulting innovations and value of this study:

Line 146-166: "While geostationary satellites enable continuous daytime observations of total VCD of $NO_2$, discrepancies between total VCD and NS concentrations of $NO_2$ concentrations remain a critical challenge. The weak correlation between NS $NO_2$ concentrations and satellite-derived total VCD of $NO_2$ (Lamsal et al., 2014) is closely tied to differences in their vertical distribution, atmospheric lifetimes, and chemical reaction pathways (Xing et al., 2017). Despite extensive efforts to derive NS $NO_2$ concentrations from total VCD measurements (Chang et al., 2025; Wei et al., 2022; Zhang et al., 2022; Dou et al., 2021), the dynamic complexity of the planetary boundary layer introduces substantial uncertainties. Moreover, prior studies have emphasized the roles of vertical $NO_2$ distribution (Sun et al., 2023; Zhang et al., 2023; Kang et al., 2021) and regional pollutant transport contributions (Yin et al., 2025; Dong et al., 2020; Chang et al., 2019; Li et al., 2017), but research explicitly linking regional transport processes to vertical $NO_2$ concentration gradients and elucidating their interactive effects remains limited. Song et al. (2024) obtained NS $NO_2$ concentrations based on the Himawari-8 geostationary satellite using machine learning, which has good performance in the noon and afternoon, and relatively poor performance in the morning. These knowledge gaps are further exacerbated by satellite data limitations in resolving NS pollution, which has direct implications for human health assessments. To address these challenges, we need to integrate ground-based remote sensing observations with in situ NS $NO_2$ measurements to investigate vertical decoupling phenomena, and investigate the influence of distinct pollutant transport pathways on NS $NO_2$ pollution dynamics."

(4) The manuscript presents an analysis of the TC/NS NO2 ratio, but its physical significance and relevance to satellite-based NO2 applications remain unclear. Besides, the sensitivity of the ratio to meteorological factors is not quantified. To strengthen the analysis, the authors should clarify the rationale for using this ratio, ensure a consistent interpretation, assess meteorological influences, and explicitly connect their findings to the use of satellite NO2 data for estimating near-surface NO2

**Response:** Thanks for your comment. We apologize for any confusion caused by our unclear statement. From the analysis of the two cases, we observe a high correlation coefficient R,

suggesting that the relationship between total VCD and NS $NO_2$ concentrations can be characterized by $y = ax + b$. However, in a larger number of cases, the correlation is less ideal, particularly during the morning hours. The low correlation does not quantify the changes between total VCD and NS, prompting us to introduce the Ratio as a means of quantification. Furthermore, the Ratio can also express the degree of dispersion between total VCD and NS to a certain extent; that is, the more unstable the Ratio, the higher the dispersion and the poorer the correlation. In contrast, the Ratio is generally more stable after 13:00, indicating the feasibility of using polar-orbiting satellites to predict NS $NO_2$ based on total VCD of $NO_2$ after this time. However, using geostationary satellites for the same prediction based on total VCD observations before 13:00 poses greater challenges. Notably, the analysis of case 1 and case 2 has already shown that regional pollution transport paths may be helpful for predicting NS $NO_2$ in winter in Beijing. In other words, considering both the spatial distribution of pollutants and their transport direction in predictions has the potential to enhance the ability of satellites to predict NS $NO_2$ concentrations based on total VCD of $NO_2$. The limitation of our study is that it is based on only one month of observations, and during this period, the geostationary satellite GEO-KOMPSAT-2B /GEMS did not release any products, which prevents us from better demonstrating the universality and applicability of our findings. In future research, we will obtain more observations and utilize geostationary satellite observation products to further refine and validate our discoveries.

We further elaborated on the role of Ratio in the MS as follows:

Line 706-712: "The Ratio, defined as the ratio of total to NS $NO_2$ concentrations, serves a dual purpose: it not only quantifies the changes between total VCD and NS concentrations of $NO_2$ when the correlation is low but also reflects the degree of dispersion between the two measurements. A more variable Ratio indicates higher dispersion and poorer correlation, providing a straightforward yet effective way to assess the reliability of using total VCD of $NO_2$ to predict NS $NO_2$ concentrations."

We further summarized the comprehensive point presented by the analysis of Ratio and the two cases as follows:

Line 737-749: "Generally, the temporal stability of the Ratio is important. The Ratio is overall less variable after 13:00, suggesting that polar-orbiting satellites can be used to predict NS $NO_2$ based on total VCD of $NO_2$ during this period with greater confidence. This temporal stability is particularly valuable because it offers a feasible approach for air quality monitoring and forecasting. In contrast, the Ratio is less stable before 13:00, posing greater challenges for using geostationary satellites for the same prediction task. It's worth noting that our analysis in winter in Beijing suggests that considering both the spatial distribution of pollutants and their transport direction has the potential to enhance the ability of satellites to predict NS $NO_2$ concentrations based on total VCD of $NO_2$. By incorporating this information into prediction models, the accuracy and reliability of satellite-based air quality predictions may be improved, particularly in complex urban environments where pollutant concentrations can vary significantly over short distances and time periods."

Further, the reviewer asks for "sensitivity of the ratio to meteorological factors is not quantified" and "assess meteorological influences". Obviously, a case study for a limited period of time, including a period when the meteorological situation did not allow for collecting the full set of data (period 2, due to the occurrence of clouds), is not suited for such analysis which requires a large variety of meteorological situations. Rather, we have identified the problem based on the analysis of a comprehensive set of data including ground-based in situ and remote sensing, satellite remote

sensing, air mass trajectory analysis and meteorological observations, which are usually not available form the same site. Furthermore, as explained in the response to comment 6 for reviewer #1, satellite data were not available during the time of the field study and thus "explicitly connect their findings to the use of satellite $NO_2$ data for estimating near-surface $NO_2$" is not possible. We regret that we are currently unable to utilize meteorological factors for quantitative assessment of Ratio in this study. In our future research, we will focus on this aspect, with the aim of establishing a connection between Ratio and satellite-observed total VCD of $NO_2$ concentrations, and making practical contributions to near-surface $NO_2$ predictions.

Specific comments:

(1) Lines 58-60: The phrase 'providing vertical total column and tropospheric densities' is unclear. It would be more precise to explicitly distinguish total column density, stratospheric column density, total and tropospheric column density. In this manuscript, 'TC' should consistently and explicitly refer to the 'tropospheric vertical column density' to avoid ambiguity.

**Response:** Thanks for your comment. We modified this sentence as follows:

Line 57-59: "Concentrations of $NO_2$ in the atmosphere can be measured using satellite-based sensors providing total and tropospheric column densities, ground-based remote sensing using MAX-DOAS or Pandora instruments, or in situ instruments."

Additionally, in this study, total VCDs refers to total column, which is derived from the nvs3 product of Pandora, while partial column $NO_2$ is obtained from the uvh3 product. We clarify this in the original text as follows:

Line 224-225: "In view of this high precision, we use total VCD of $NO_2$ from the nvs3 product in this study and select data with quality control flag of L10."

Line 237-239: "The $NO_2$ of the partial column can be obtained from the uvh3 product which was downloaded from the PGN website (https://pandonia-global-network.org, last accessed: 22 Jan 2025)."

(2) Lines 85-88: Thompson's work focuses on the complexity of the TC-NS NO2 relationship, which does not align well with the paragraph's main discussion on Pandora vs. satellite TC NO2 This reference would be more appropriate in a section specifically addressing TC-NS variability rather than in a discussion of measurement consistency between Pandora and satellite data.

**Response:** Thanks for your comments. We have moved Thompson's work to the next paragraph to discuss the complex relationship between total VCD and NS $NO_2$.

Line 113-119: "… Similarly, Liu et al. (2024) show different relations between total VCD and NS concentrations of $NO_2$ for low and high concentrations which are qualitatively explained in terms of transport and local emissions. Moreover, Thompson et al. (2019), using data from the KORUS-AQ coastal cruise experiment, reported that there is no consistent correlation between total VCD and NS concentrations of $NO_2$ across different cases and that the relation between total VCD and NS concentrations of $NO_2$ is complex."

(3) Lines 109-114: This sentence is excessively long, making it difficult to follow. It is recommended to split it into two sentences to improve readability and ensure clarity of the message.

**Response:** Thanks for your comment. We have modified this sentence as follows:

Line 107-113: "Their results indicate that total VCD and NS concentrations of $NO_2$ exhibit a stronger correlation under advective boundary layer conditions at high wind speeds, where the vertical distribution of $NO_2$ is more uniform. In contrast, in the presence of plumes from large point sources, either decoupled from the surface or transported from nearby cities, enhance the vertical heterogeneity of $NO_2$. These plumes contribute to a less consistent relationship between total VCD and NS concentrations of $NO_2$."

(4) Figure 2: It is suggested to add dashed lines in Figure 2 to clearly indicate the boundaries between the three periods, improving readability.
**Response:** Thanks for your comment. We have added the boundaries between the three periods using dash lines in the Figure 2.

[Figure]

**Figure 2**. Time series of observed parameters from Jan 10 to 29, 2022 (a) total VCD and NS concentrations of $NO_2$ concentrations, (b) NS $PM_{2.5}$ concentration, (c) temperature and related humidity, and (d) wind speed and wind direction from WMO meteorological station 54511 in Beijing. The vertical dotted lines mark the boundaries between the three periods and the yellow shaded rectangles mark the two cases discussed in Section 3.2.

(5) Lines 315-322: The manuscript presents PM2.5 observations and results but does not explicitly explain their relevance to NO2 analysis. To improve clarity, the authors should justify the inclusion of PM2.5 data and clarify how it supports the study's objectives, ensuring a stronger connection between NO2 assessment and aerosol pollution.
**Response:** Thanks for your comment. The $PM_{2.5}$ mass concentration observations serves two purposes: (1) NS pollution in the Beijing area is usually not dominated by $NO_2$, but rather by $PM_{2.5}$. Also, the air mass trajectories for case 2 the increased wind speed at 16:00, which marks the end of the pollution episodes. In fig 2 we see that clearly in both the $NO_2$ and $PM_{2.5}$ concentrations, so we

use $PM_{2.5}$ as an additional indicator. (2) Since our vertical information comes from Lidar, and the Lidar signal is caused by the backscattering from particulate matter, the NS $PM_{2.5}$ mass concentration serves as auxiliary information for Lidar observations. Therefore, we believe that the $PM_{2.5}$ observations is necessary. We have also utilized $PM_{2.5}$ information on multiple occasions, for example:

Line 495-497: "The increase of the NS concentrations is consistent with the highest $PM_{2.5}$ concentrations as presented in Fig. 2b and the overall increase of the lidar signal, indicating increasing aerosol concentrations."

Line 606-608: "The lidar data in Fig. 5c, with lower intensity than on 14 January, indicate smaller aerosol concentrations on 18 January than on 14 January, consistent with the smaller $PM_{2.5}$ concentrations (in Fig. 2b)."

(6) Line 484: The TC/NS ratio mentioned in Line 484 is not found in Figure 3b. Please clarify whether it is missing or referenced incorrectly.

**Response:** Thanks for your comment. We apologize for this mistake. Instead of "ratio," it should be "relationship." We have revised this sentence as follows:

Line 525-528: "As a result, the temporal variation of the concentrations in both layers was in part influenced by the same processes, differences were not large (Fig. 3a) and the relationship between the total VCD and NS concentrations was linear with a small slope (174.24) and well-correlated (R=0.94) (Fig. 3b)."

(7) Figure 3: There appear to be unexpected content between Figures 3c and 3d, likely due to figure cropping. Please check and correct this issue.

**Response:** Thanks for your comment. We have updated the figure 3 and 5 as follows:

[Figure]

**Figure 3.** (a) Time series of NS $NO_2$ (grey line) and total VCD of $NO_2$ (blue circles) at the Beijing

RADI site (40.004°N, 116.379°E) on Jan 14, 2022; (b) scatterplots of total VCD and NS concentrations of $NO_2$ and fits to these data during the morning (before 13:00) and during the afternoon (after 13:00), showing different relationships as discussed in the text; (c) time series of vertical profiles of range-corrected lidar signal at 1064 nm. Note that the lowest height in Fig. 3c is 100 m; (d) time series of $NO_2$ vertical profiles derived from Pandora sky radiance measurements. Note that the Pandora profiles are constructed from layer-averaged volume mixing ratios interpolated to 6 standard levels and the lowest level is 0.2 km.

[Figure]

**Figure 5**. Same as Fig. 3 but for 18 January, 2022.

(8) Figure 4: To enhance clarity, it is suggested to add necessary subtitles for the three subfigures in Figure 4c, such as local time. Additionally, the resolution of Figure 4 appears low and should be improved. (Same for Figure 6)

**Response:** Thanks for your comment. We have added the subtitles for the subfigures and improved the resolution of Figure 4 as follows:

[Figure]

**Figure 4.** (a) Spatial distribution of tropospheric NO$_2$ in the study area derived from TROPOMI data on 14 January 2022; (b) Synoptic weather map at 00:00 UTC (08:00 LT); 24-hour backward air mass trajectories arriving at the Beijing-RADI site at (c) 10:00, (d) 13:00 and (e) 16:00 LT, at heights of 300, 100 and 1000 m, calculated using the HYSPLIT model with 6h time steps (00, 06, 12 and 18) and a shorter time step to the arrival time.

(9) Figure 6: The middle panel of Figure 6c contains unexpected content. Please verify and correct this issue.

**Response:** Thanks for your comment. We have added the subtitles for the subfigures and updated the Figure 6 as follows:

[Figure]

**Figure 6**. Same as Fig. 4, but for 18 January, 2022.

(10) Lines 691-693: The outlook on potential large-scale implementation is vague. The authors should specify which shortcomings to be resolved.

**Response:** Thanks for your comment. We further pointed out the limitations of the study and revised the manuscript as follows:

Line 796-799: "Moreover, we will broaden the scope of experimental areas and field sites to complement research on the various pollutant emission and transport characteristics. Furthermore, observations over a longer period will allow us to capture more representative cases, thereby enhancing the reliability of our findings."

---

## Author Comment (AC3)

**Reviewer #2:**

The authors have effectively addressed my previous comments, and I appreciate their efforts in revising the manuscript. However, I am concerned that the narrative remains overly detailed in places, which may make it difficult for readers to follow the main findings. I recommend a thorough revision to streamline the manuscript before it can be considered for publication in Atmospheric Chemistry and Physics (ACP).

For example, in lines 420-442 and 580-605 of Sections 3.2.1 and 3.2.2, the manuscript provides extensive descriptions of Figures 3 and 5, listing numerous numerical values at dense time points to illustrate the temporal changes in NS and total VCD. I suggest condensing these sections by highlighting the most important information and key messages conveyed by each figure. Highlighting representative data and key trends will help readers to better understand the key findings.

**Responds:** We have shortened the description of figures to be more accessible, as you suggested. The revisions to the manuscript are as follows:

**Line 421-429**: NS concentrations showed diurnal variation, decreasing from midnight (0.11 mg m⁻³) to morning (0.065 mg m⁻³ at 10:30), followed by an afternoon increase and strong evening fluctuations (0.04-0.175 mg m⁻³) likely associated with rush hour emissions and domestic heating. During the limited observation window (10:30-15:30) for Pandora, both VCD and NS concentrations initially showed similar behavior with minimal variation. However, their temporal patterns diverged after 13:00 while NS concentrations plateaued, VCD exhibited rapid growth (40 to 72 mg m⁻² between 12:00-15:00), nearly doubling before declining slightly.

**Line 451-464**: The lidar observations in Fig. 3c revealed a complex boundary layer structure with distinct aerosol layers. The data show a well-mixed shallow boundary layer between midnight and 03:00 and the formation of an internal boundary layer after about 04:00, disconnected from the layer above. The internal boundary layer rises gradually until about 11:00 (up to about 400 m), with the clean layer above (between 400 and 500 m), and a new layer (at heights of approximately 800 to 900 m) appears around 07:00, probably due to advection. This vertical variation indicates a disconnected boundary structure with two disconnected layers, likely caused by nocturnal cooling (Stull, 1988). The temperature gradient prohibited material exchange between these layers, leading to the accumulation of near-surface emissions and the trapping of trace gases and aerosols in the upper layer. The occurrence of such a situation is consistent with the observations discussed in Section 3.1 and Fig. 2, with low wind speed, lowest air temperature during period I (-12°C) and enhanced RH (indicating trapping of water vapor together with decreased air temperature).

**Line 564-575**: The observational data for 18 January reveal distinct diurnal patterns in NO₂ dynamics, with NS concentrations exhibiting higher baseline levels than on 14 January while following a similar initial decreasing trend until 11:30. Subsequently, NS concentrations demonstrated nonlinear growth, plateauing at 0.12 mg m⁻³ (about 1 hour after 14:30) before further increasing to 0.16 mg m⁻³ by 21:00, attributed to combined rush-hour emissions and reduced photochemical dissipation. Concurrently, total vertical column density (VCD) displayed accelerated

morning depletion (from 54 to 36 mg m$^{-2}$ between 08:30 and 12:30), but in contrast to the 14$^{th}$, after 11:30 the total VCD of NO$_2$ concentrations continued to decrease while the NS NO$_2$ concentrations increased. Hence, in this situation, it may be difficult to determine NS NO$_2$ concentrations from the relationship (R=0.40) before 13:00 (Fig 5b).

---

## Author Response (AR2)

1. Line 617: Please spell out "WNW" in full, consistent with the use of "southwest (SW)" and "southeast (SE)" in Line 405.

**Response:** We have been modified the it as "This can be explained by the transport from clean areas to the **west and west-north-west** of Beijing, as indicated by the air mass trajectories arriving in Beijing at 300m, 500 m and 1000m, at 10:00, 13:00 and 16:00 LT (Fig. 6c)."

2. Line 628: "12:00 LT" should be corrected to "13:00 LT"

**Response:** Corrected. Thanks for pointing this out.

3. Lines 679–680: Please revise "The Ratio, defined as the ratio of total to NS $NO_2$ concentrations" to "The Ratio, defined as the ratio of total VCD to NS $NO_2$ concentrations," as "VCD" is missing.

**Response:** Corrected. Thanks for pointing this out.